# Altered function and differentiation of age-associated B cells contribute to the female bias in lupus mice

Edd Ricker[1,2], Michela Manni[1], Danny Flores-Castro[1], Daniel Jenkins[1], Sanjay Gupta[1], Juan Rivera-Correa[1,2], Wenzhao Meng[3], Aaron M. Rosenfeld [3], Tania Pannellini[4], Mahesh Bachu[5], Yurii Chinenov[6], Peter K. Sculco[7], Rolf Jessberger [8], Eline T. Luning Prak [3] & Alessandra B. Pernis [1,2,6,9✉]

Differences in immune responses to viruses and autoimmune diseases such as systemic lupus erythematosus (SLE) can show sexual dimorphism. Age-associated B cells (ABC) are a population of CD11c[+]T-bet[+] B cells critical for antiviral responses and autoimmune disorders. Absence of DEF6 and SWAP-70, two homologous guanine exchange factors, in double-knock-out (DKO) mice leads to a lupus-like syndrome in females marked by accumulation of ABCs. Here we demonstrate that DKO ABCs show sex-specific differences in cell number, upregulation of an ISG signature, and further differentiation. DKO ABCs undergo oligoclonal expansion and differentiate into both CD11c[+] and CD11c[−] effector B cell populations with pathogenic and pro-inflammatory function as demonstrated by BCR sequencing and fate-mapping experiments. *Tlr7* duplication in DKO males overrides the sex-bias and further augments the dissemination and pathogenicity of ABCs, resulting in severe pulmonary inflammation and early mortality. Thus, sexual dimorphism shapes the expansion, function and differentiation of ABCs that accompanies TLR7-driven immunopathogenesis.

[1] Autoimmunity and Inflammation Program, Hospital for Special Surgery, New York, NY, USA. [2] Department of Microbiology and Immunology, Weill Cornell Medicine, New York, NY, USA. [3] Department of Pathology and Laboratory Medicine, Perelman School of Medicine, Philadelphia, PA, USA. [4] Research Division and Precision Medicine Laboratory, Hospital for Special Surgery, New York, NY, USA. [5] Arthritis and Tissue Degeneration Program, Hospital for Special Surgery, New York, NY, USA. [6] David Z. Rosensweig Genomics Research Center, Hospital for Special Surgery, New York, NY, USA. [7] Department of Orthopedic Surgery, Hospital for Special Surgery, New York, NY, USA. [8] Institute of Physiological Chemistry, Technische Universitat, Dresden, Germany. [9] Department of Medicine, Weill Cornell Medicine, New York, NY, USA. ✉email: pernisa@hss.edu

Sex-dependent differences in immune responses have been well documented in viral infections, vaccination outcomes, and autoimmune diseases like Systemic Lupus Erythematosus (SLE), a heterogeneous disorder that often includes upregulation of interferon stimulated genes (ISG) in addition to autoantibody production and multi-organ involvement[1,2]. Both sex hormones and genes on the X-chromosome have been implicated in this sexual dimorphism[3]. Notably, TLR7, an endosomal TLR critical for responses to viruses like SARS-CoV-2 and lupus pathogenesis, is encoded on the X-chromosome and has recently been shown to partially escape X-chromosome inactivation resulting in greater TLR7 expression in a proportion of female B cells, monocytes, and pDCs[4].

TLR7 engagement promotes the formation of Age/autoimmune-associated B cells (ABC), also known as DN2 in humans, a B cell population that preferentially expands with age in female mice[5–7]. ABCs exhibit a distinctive phenotype and, in addition to classical B cell markers, express the transcription factor T-bet and myeloid markers like CD11c. T-bet and CD11c are often, but not always, co-expressed[8–10]. ABCs are an important component of antiviral responses and are inappropriately controlled in several viral infections including HIV and SARS-Co-V2[11–13]. Aberrant expansion and activation of ABCs is also associated with autoimmune pathogenesis, especially SLE[14]. In this disease, ABCs accumulate to a greater extent in African-American patients, rapidly differentiate into plasmablasts/plasma cells (PB/PC), are major producers of autoantibodies, and correlate with disease activity and clinical manifestations[7,15,16]. Despite the emerging biological and clinical importance of ABCs, the full spectrum of their function and differentiation capabilities are incompletely understood.

Although T-bet is a well-known marker for ABCs, reliance of ABCs on this transcription factor differs depending on the setting. B-cell T-bet is important for protective flu-specific IgG2c antibodies, but its absence has variably impacted the generation of ABCs and disease parameters in lupus murine models[9,17–19]. This is likely due to the presence of additional regulators of autoimmune ABCs such as IRF5 (Interferon Regulatory Factor 5), whose dysregulation promotes ABC accumulation and lupus development[20]. The ability of IRF5 to drive ABC expansion can be restrained by the SWEF proteins, Def6 and SWAP-70, two homologous proteins that also control cytoskeletal reorganization by regulating Rho GTPase signaling and whose combined absence in mice results in a lupus syndrome that primarily affects females[21–24]. An important role for these molecules in immune responses, inflammation, and autoimmunity is supported not only by murine genetic models but also by human studies. The CORO1A-DEF6 blood transcription module correlates with responses to flu vaccination and malaria[25,26]. Furthermore, SWAP70 is a susceptibility locus for RA[27] and CVD[28] while DEF6 is a risk factor for human SLE[29,30]. Mutations in DEF6 moreover result in early-onset autoimmune manifestations, often associated with viral infections, which include autoantibody production and upregulation of an ISG signature[31,32].

In this study we have exploited the sex-bias exhibited by mice lacking both SWEF proteins, DEF6 and SWAP-70 (Double-Knock-out or DKO) to investigate the impact of sexual dimorphism on ABC function. We show that ABCs from DKO females and males differ in their ability to expand, upregulate an ISG signature, and further differentiate. BCR sequencing and fate mapping experiments indicate marked oligoclonal expansion and interrelatedness of ABCs with both CD11c+ and CD11c− effector populations, which include CD11c+ pre-GC B cells and CD11c+ PBs. In addition to IRF5, DKO ABCs also require IRF8 but are less dependent on T-bet. Notably, Tlr7 duplication in DKO males overrides the sex-bias and augments the pathogenicity of ABCs

resulting in severe pathology and early mortality. Thus, in autoimmune settings, ABCs can give rise to a heterogenous population of effector cells with distinct pathogenic potentials that are controlled in a sexually dimorphic manner.

## Results

### ABC accumulation and function in DKOs is sex-dependent and controlled by TLR7.
Similar to human SLE, the lupus syndrome that develops in DKOs preferentially affects females providing a powerful model to delineate the cellular and molecular mechanisms that underlie sexual dimorphism in autoimmunity. Given the key role of ABCs in lupus, we first assessed whether the sex-bias that accompanies lupus development in DKOs was associated with differences in ABC expansion. Significantly more ABCs accumulated in DKO females than age-matched DKO males, although DKO males still contained greater numbers of ABCs than WT controls (Fig. 1a; Supplementary Fig. 1A). Furthermore, ABCs sorted from DKO males secreted significantly lower levels of anti-dsDNA IgG2c upon stimulation with a TLR7 agonist, imiquimod, than ABCs from DKO females (Fig. 1b). Thus, both the accumulation and the function of ABCs in DKOs are controlled in a sex-specific manner.

Tlr7 can be expressed biallelically in a proportion of female B cells due to incomplete X chromosome inactivation[4]. In line with these findings, ABCs from DKO females expressed higher levels of Tlr7 than ABCs from DKO males (Supplementary Fig. 1B). ABC accumulation in DKO females was furthermore dependent on TLR7, as DKO females crossed to Tlr7−/− mice exhibited a profound reduction in ABC accumulation (Fig. 1c). Tlr7-deficient DKO females also displayed significant decreases in GC B cells, CD138+ plasmablasts/plasma cells (PB/PCs), and T_FH cells, and lacked anti-dsDNA IgG2c antibodies (Supplementary Fig. 1C–F). Thus, ABC expansion and lupus pathogenesis in DKO females are dependent on Tlr7.

To further assess the importance of Tlr7 in the sex-bias of DKOs, we crossed DKO males to C57BL/6 mice carrying the Y-linked genomic modifier Yaa (termed Yaa-DKOs), in which a portion of the X-chromosome has translocated onto the Y-chromosome resulting in a 2-fold increase in Tlr7 expression in males[33]. Tlr7 duplication in DKO males markedly increased the frequencies and numbers of splenic ABCs reaching levels that were even greater than those observed in DKO females (Fig. 1d; Supplementary Fig. 1F). Tlr7 duplication in DKO males also rescued the ability of sorted male ABCs to secrete anti-dsDNA IgG2c antibodies upon stimulation (Fig. 1e). Increased ABC accumulation and function in Yaa-DKO males were accompanied by autoantibody production, the classical clinical feature of SLE (Fig. 1f, g). Total antibody titers were also comparable between DKO females and Yaa-DKO males (Supplementary Fig. 1H). Yaa-DKO males also exhibited significantly decreased survival as compared to both DKO males and females (Fig. 1h). Thus, duplication of Tlr7 in Yaa-DKO males overrides the sex-bias and promotes the development of a severe lupus syndrome in DKO males marked by greatly enhanced accumulation of ABCs and autoantibody responses.

### The expansion of GC B cells and PB/PCs in DKOs is regulated in a sex-specific manner.
In addition to ABC accumulation, DKO females also exhibit robust GC and PB/PC responses[22], prompting us to examine whether sex-specific differences could also be observed in these compartments. DKO females contained more GL7+Fas+ GC B cells than DKO males, a difference that was again reversed by Tlr7 duplication in Yaa-DKO males (Fig. 2a; Supplementary Fig. 2A). Immunofluorescence staining confirmed these findings and revealed that GCs in Yaa-DKO

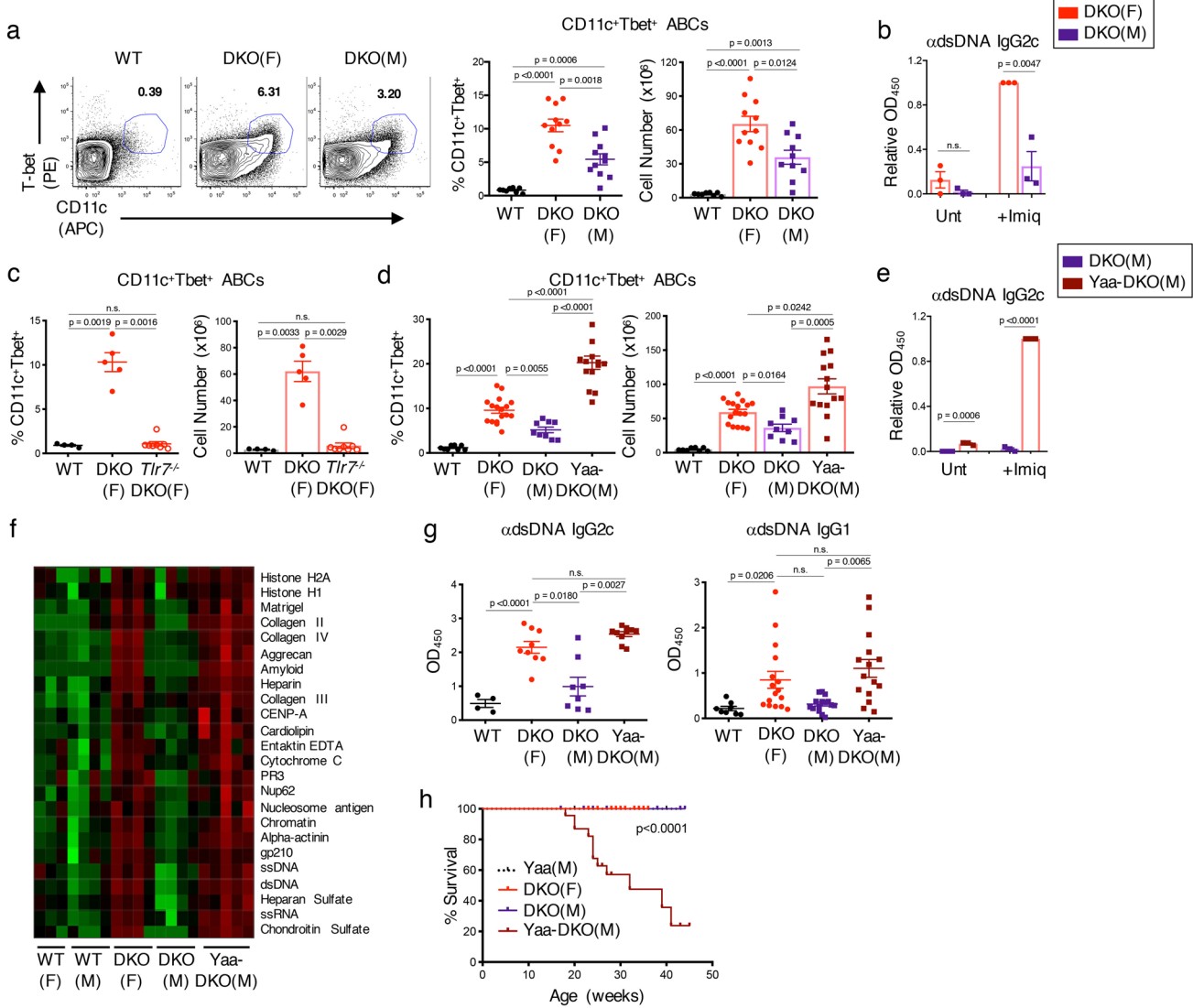

**Fig. 1 TLR7 controls sex-specific differences in ABC formation and function. a** Representative FACS plots and quantifications of CD11c[+]Tbet[+] ABCs (gated on B220[+]) from spleens of aged (24+wk) female C57BL/6 (WT) *(black circles)*, female DKO (F) (red circles), and male DKO (M) mice (purple squares). Data shows mean ± SEM; *n* = 8 for WT, *n* = 11 for DKO(F), *n* = 10 for DKO(M) over 8 independent experiments; *p*-value by Brown–Forsythe and Welch ANOVA followed by Games–Howell's test for multiple comparisons. **b** Pooled ELISA data for anti-dsDNA IgG2c antibodies from supernatants of sorted ABCs (B220[+]CD19[+]CD11c[+]CD11b[+]) from spleens of aged (24+wk) DKO(F) and DKO(M) mice after culturing with Imiquimod for 7d. Data shows mean ± SEM; *n* = 3; *p*-value by unpaired two-tailed *t*-tests. **c** Quantifications of CD11c[+]Tbet[+] ABCs from the spleens of aged (24+wk) female WT, DKO(F), and female *Tlr7*-deficient DKO (*Tlr7*[−/−].DKO(F)) mice (red open circles). Data show mean ± SEM; *n* = 4 for WT, *n* = 5 for DKO(F), *n* = 8 for *Tlr7*[−/−].DKO (F) mice over 4 independent experiments; *p*-value by Brown–Forsythe and Welch ANOVA followed by Games–Howell's test for multiple comparisons. **d** Quantifications of CD11c[+]Tbet[+] ABCs (gated on B220[+]) from the spleens of aged (24+wk) WT, DKO(F), DKO(M), and male Yaa-DKO(M) mice (maroon squares). Data shows mean ± SEM; *n* = 8 for WT, *n* = 17 for DKO(F), *n* = 9 for DKO(M), *n* = 14 for Yaa-DKO(M) over 8 independent experiments; *p*-value by Brown–Forsythe and Welch ANOVA followed by Games–Howell's test for multiple comparisons. **e** Pooled ELISA data for anti-dsDNA IgG2c antibodies from supernatants of sorted ABCs from aged (24+wk) DKO(M) or Yaa-DKO(M) mice after culturing with Imiquimod for 7d. Data shows mean ± SEM; *n* = 3; *p*-value by unpaired two-tailed *t*-tests. **f** Autoantigen microarray showing the relative autoantibody levels in the serum of aged (24+wk) female WT (F), male WT (M), DKO(F), DKO(M), and Yaa-DKO(M) mice; *n* = 3 for WT(F), *n* = 4 for WT(M), DKO(F), and DKO(M), *n* = 5 for Yaa-DKO(M). **g** Pooled ELISA data for anti-dsDNA IgG2c and IgG1 in the serum from the indicated aged (24+wk) mice. Data shows mean ± SEM; for anti-dsDNA IgG2c, *n* = 4 for WT, *n* = 9 for DKO(F), *n* = 8 for DKO(M), *n* = 10 for Yaa-DKO(M); for anti-dsDNA IgG1, *n* = 8 for WT, *n* = 16 for DKO(F), *n* = 14 for DKO(M); *n* = 15 for Yaa-DKO(M); *p*-value by Brown–Forsythe and Welch ANOVA followed by Games–Howell's test for multiple comparisons. **h** Plot showing survival rate of Yaa control, DKO(F), DKO(M), and Yaa-DKO(M) mice across 45 weeks. Data represents a cohort of 14 Yaa control, 16 DKO (F), 9 DKO(M), and 31 Yaa-DKO(M) mice; *p*-value by Mantel–Cox test.

males were smaller and less well-organized than those in DKO females (Fig. 2b; Supplementary Fig. 2B). DKO females also demonstrated a greater expansion of PB/PCs than DKO males (Fig. 2c). *Tlr7* duplication in Yaa-DKO males reversed this effect and strongly promoted the accumulation of PB/PCs, which was

primarily observed in the spleen but not in the BM (Fig. 2c; Supplementary Fig. 2C). No sex-based differences were detected in other B cell compartments (Supplementary Fig. 2D–G). Other parameters known to promote spontaneous GC responses such as the ratio between $T_{FH}$ and $T_{FR}$ or the dual production of IFNγ

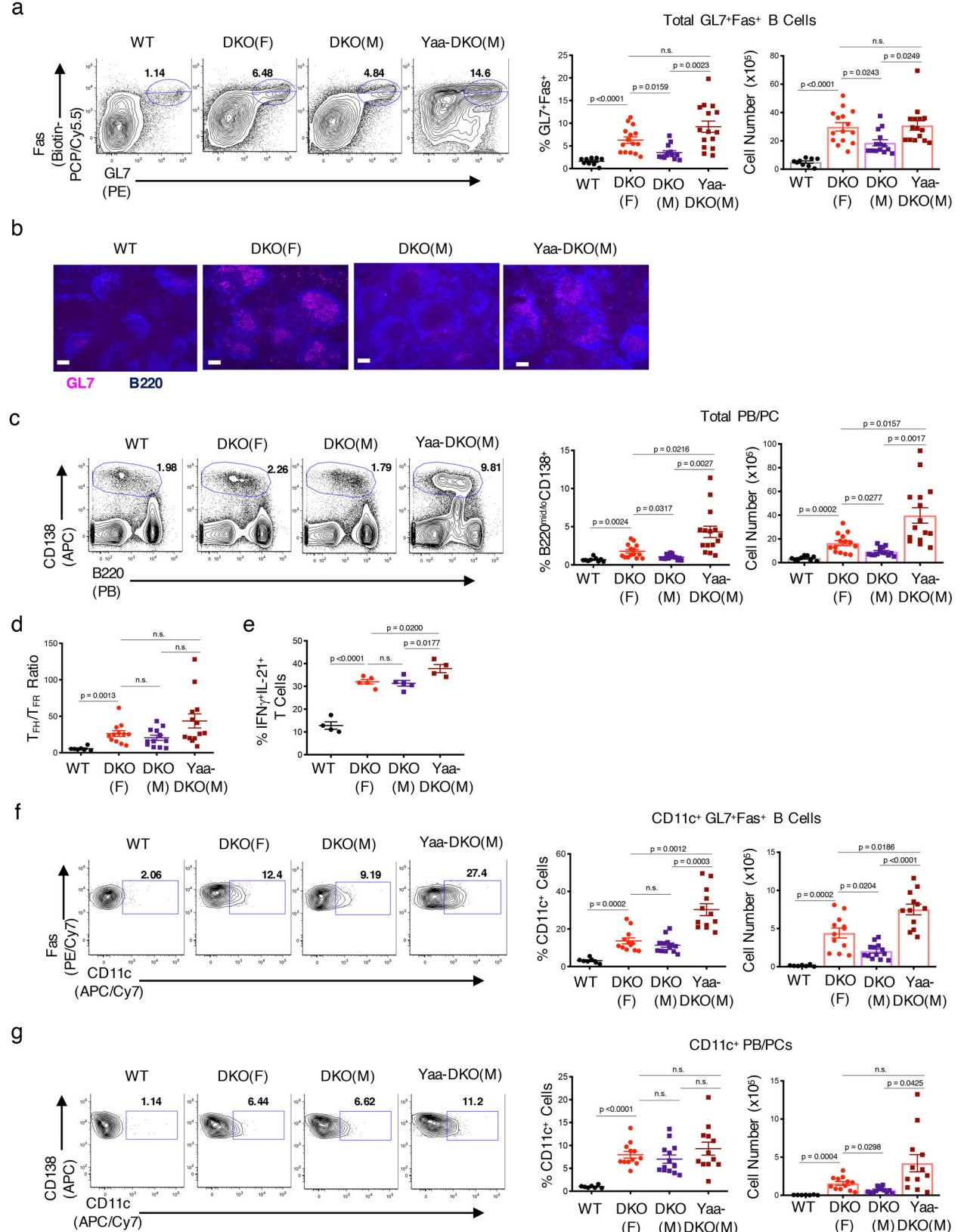

and IL-21 were comparable between DKO females and males and only minimally affected by *Tlr7* duplication (Fig. 2d, e; Supplementary Fig. 2H–J). Thus, DKOs exhibit a sex-specific accumulation of GC B cells and PB/PCs, which can be regulated in a *Tlr7*-dependent manner.

Given the sex-bias in the accumulation of ABCs as well as of GC B cells and PB/PCs, we next investigated whether these populations might be related. An analysis of the GC B cell population in DKO females revealed that a subset of these cells expressed CD11c and that the numbers of CD11c$^+$ GC B cells

**Fig. 2 Sex-dependent accumulation of GC B cells and PB/PCs in DKO mice. a** Representative FACS plots and quantifications of total GC B cells (gated on B220$^+$ GL7$^+$Fas$^+$ splenocytes) from aged (24+wk) female C57BL/6 (WT) (black circles), female DKO(F) (red circles), male DKO(M) (purple squares), and male Yaa-DKO(M) mice (maroon squares). Data show mean ± SEM; $n = 9$ for WT, $n = 15$ for DKO(F) and Yaa-DKO(M), $n = 14$ for DKO(M) over 9 independent experiments; $p$-value by Brown–Forsythe and Welch ANOVA followed by Games–Howell's test for multiple comparisons. **b** Representative immunofluorescence images of B220$^+$ (blue) GL7$^+$ (pink) GCs in spleens from the indicated mice. Data representative of at least 4 frames for at least 2 mice per genotype. Bars show 50 µm. **c** Representative FACS plots and quantifications of total PB/PC (B220$^{mid/lo}$CD138$^+$) in spleens of the indicated mice. Data show mean ± SEM; $n = 9$ for WT, $n = 14$ for DKO(F) and DKO(M), $n = 15$ for Yaa-DKO(M) over 9 independent experiments; $p$-value by Brown–Forsythe and Welch ANOVA followed by Games–Howell's test for multiple comparisons. **d** Quantification of the FACS plots showing T$_{FH}$ (CD4$^+$ PD1$^{hi}$CXCR5$^+$ Foxp3$^-$) to T$_{FR}$ (CD4$^+$ PD1$^{hi}$CXCR5$^+$ Foxp3$^+$) ratio in spleens from the indicated mice. Data show mean ± SEM; $n = 7$ for WT, $n = 12$ for DKO(F) and DKO(M), $n = 13$ for Yaa-DKO(M) over 7 independent experiments; $p$-value by Brown–Forsythe and Welch ANOVA followed by Games–Howell's test for multiple comparisons. **e** Quantification of FACS plots showing IFNγ$^+$IL-21$^+$ T cells in spleens of the indicated mice following 4 h treatment of splenocytes with PMA/Ionomycin. Data show mean ± SEM; $n = 4$ for WT and Yaa-DKO(M), $n = 5$ for DKO(F) and DKO(M) over 4 independent experiments; $p$-value by Brown–Forsythe and Welch ANOVA followed by Games–Howell's test for multiple comparisons. **f, g** Representative FACS plots and quantifications of CD11c$^+$ GL7$^+$Fas$^+$ cells (gated on B220$^+$ splenocytes) (**f**) or CD11c$^+$ PB/PC (B220$^{mid/lo}$CD138$^+$) (**g**) from the indicated mice. Data show mean ± SEM; $n = 7$ for WT, $n = 12$ for DKO(F) and Yaa-DKO(M), $n = 3$ for DKO(M) over 7 independent experiments; $p$-value by Brown–Forsythe and Welch ANOVA followed by Games–Howell's test for multiple comparisons.

were significantly greater in DKO females than DKO males or WT controls (Fig. 2f; Supplementary Fig. 2K). CD11c$^+$ GC B cells were greatly increased in Yaa-DKO males (Fig. 2f). We also identified a population of CD11c$^+$ PB/PCs that accumulated in DKO females and, to an even greater extent, in Yaa-DKO males (Fig. 2g; Supplementary Fig. 2L). Thus, sex differences in lupus development in DKOs are accompanied by the aberrant accumulation of CD11c-expressing B cell effector subsets.

**Oligoclonal expansion and interrelatedness of ABCs and CD11c$^+$ and CD11c$^-$ B cell effector populations**. To gain further insights into the effector B cell subsets that differentially expand in the spleens of DKO mice, we next compared their BCR repertoires. To evaluate the clonal landscape, we began by determining the contribution of the top 20 ranked clones to the overall repertoire by computing the D20 index[34]. We observed that the D20 index, or fraction of sequence copies contributing to the sum of the top 20 ranked clones, was lowest in follicular B cells (FoBs) and increased in ABCs, followed by GCB and finally being highest (most expanded) in the PB/PC pool (Fig. 3a). This order of large clone contribution by B cell subset was preserved in both DKO females and Yaa-DKO males (Supplementary Fig. 3A). Furthermore, when one studies the level of resampling of clones between replicate sequencing libraries as an independent measure of clone size, the same trend is preserved, with FoBs having the lowest degree of overlap and PB/PCs having the highest (Fig. 3b; Supplementary Fig. 3B). We next analyzed the level of somatic hypermutation (SHM), which revealed that GCB and PB/PC fractions had the highest frequencies of clones with SHM, while the ABCs had a level of SHM that was intermediate between FoBs and GCB/PBs (Fig. 3c; Supplementary Fig. 3C, D). The SHM distribution trended by B cell subset rather than by mouse strain and the relative levels of SHM were preserved across these different B cell subsets irrespective of whether clones were unweighted or weighted by size (Fig. 3d; Supplementary Fig. 3C, D). Given the somewhat lower levels of SHM in the ABCs as compared to the other B cell populations, we next turned to the length of the third complementary determining region (CDR3) and heavy chain variable (VH) gene usage as other general repertoire features. This analysis revealed longer CDR3 lengths for FoBs compared to the other B cell populations and VH gene usage that differed between FoBs and the other subsets (Fig. 3e, f). Taken together, this initial global repertoire analysis revealed that general repertoire features tended to map by B cell subset rather than by mouse strain, with FoBs having the greatest diversity, smallest clone size, and lowest level of SHM and GCB/PBs having

the highest. ABCs instead tended to be intermediate with most of these measures.

To further assess the relationships of ABCs with the other B cell effector populations that aberrantly expand in DKO females and Yaa-DKO males, we sorted and analyzed the various populations from individual mice, stratifying the GCB and PB/PC subsets by CD11c expression. CD11c is differentially expressed on DN B cells in human lupus and we wondered if similar interconnections existed in the context of the DKO females and Yaa-DKO male models. In particular, CD11c$^+$ DN2 cells are transcriptionally and epigenetically poised to become PBs in human SLE[35]. If this also occurs in these models of murine SLE, then CD11c$^+$ ABCs, which are phenotypically and functionally overlapping with DN2, may exhibit a higher degree of clonal overlap with PB/PCs than with GCBs or with CD11c$^-$ subsets. We therefore visualized overlapping clones from DKO female and Yaa-DKO male mice as strings and in Venn Diagrams (Fig. 3g, h; Supplementary Fig. 3E–J). These visualizations revealed a high level of clonal overlap across all subsets, including hundreds of clones that were present in all the subsets. To quantify and compare the level of overlap between the different subsets, we compared the Jaccard index, in which each clone is only counted once in each subset, and the Cosine similarity, in which clone size is also taken into account (Fig. 3i; Supplementary Fig. 3J, K). Neither measure revealed a consistent pattern of similarity with respect to CD11c status. However, ABCs were consistently most highly associated with PB/PCs.

To further verify the relationships between ABCs and the other effector populations in DKOs, we crossed them with Tbet-zsGreen-T2A-CreER$^{T2}$-Rosa26-loxP-STOP-loxP-tdTomato mice (termed ZTCE-DKO), where cells expressing T-bet co-express zsGreen and can be traced with Tamoxifen-inducible tdTomato expression[36]. These mice allow for the detection of stable T-bet-expressing cells (zsGreen$^+$tdTomato$^+$) and cells that previously but no longer express T-bet (zsGreen$^-$tdTomato$^+$)[36]. ABCs expressed high levels of both ZsGreen and tdTomato, indicating stable expression of T-bet (Fig. 3j, k). CD11c$^+$ and CD11c$^-$ GC B cells contained both zsGreen$^+$tdTomato$^+$ and zsGreen$^-$tdTomato$^+$ although the CD11c$^-$ subset was preferentially zsGreen$^-$tdTomato$^+$ suggesting that both populations can originate from T-bet expressing cells but that CD11c$^+$ GC B cells include a greater fraction of stable T-bet expressors (Fig. 3j, k). Although no longer expressing ZsGreen, a substantial fraction of CD11c$^+$ PB/PCs and CD11c$^-$ PB/PCs were tdTomato$^+$, suggesting that both these populations can derive from T-bet-expressing B cells (Fig. 3j, k). In combination with our clonal overlap analyses, these findings suggest that, in this autoimmune setting, ABCs share lineage

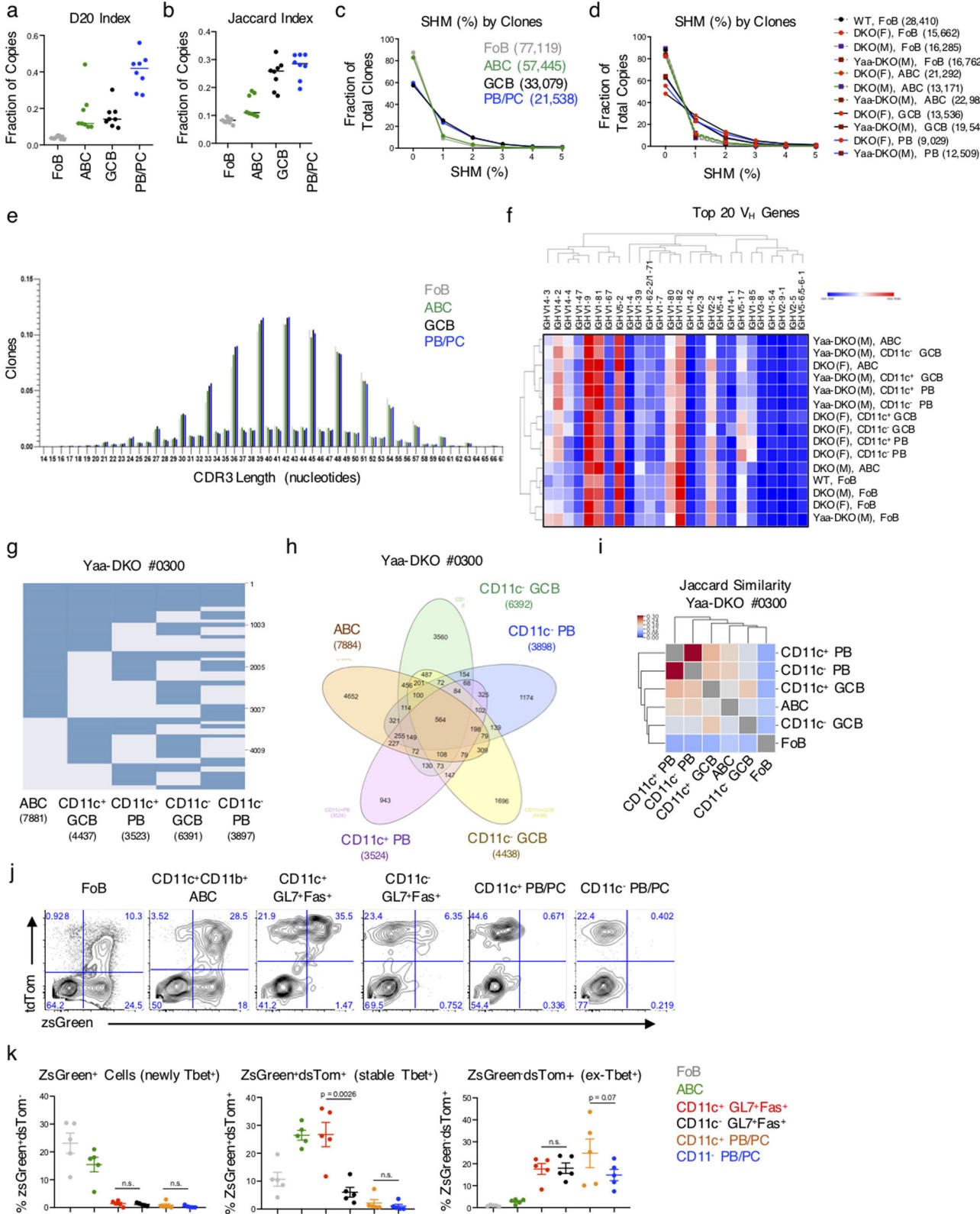

relationships with both CD11c+ and CD11c− GC B cell and PB/PC populations.

**Enrichment for an ISG signature differentiates ABCs from DKO females and males**. While ABCs from female and male DKOs exhibited similar BCR repertoire features, their marked differences in function and differentiation suggested that they might employ distinct molecular programs. To gain insights into these mechanisms, we compared their transcriptome by RNA-seq (Fig. 4a). Gene set enrichment (GSEA) and CPDB pathway analyses revealed that ABCs from DKO females were enriched for pathways related to SLE pathogenesis, interferon (IFN) responses, and TLR and complement cascades (Fig. 4b; Supplementary Fig.

**Fig. 3 B cell effector populations exhibit a high degree of interrelatedness.** IgH sequencing was performed on sorted FoBs (B220$^+$CD11c$^−$CD11b$^−$CD23$^+$) (*gray circles*), ABCs (B220$^+$CD11c$^+$CD11b$^+$) (*green circles*), CD11c$^+$ and CD11c$^−$ GL7$^+$CD38$^{lo}$ B cells (gated on B220$^+$ splenocytes) (*black circles*), and CD11c$^+$ and CD11c$^−$ PB/PCs (B220$^{lo}$CD138$^+$) (*blue circles*) from aged (24+wk) female C57BL/6 (WT), female DKO(F), male DKO(M), and male Yaa-DKO(M) mice. **a** D20 index of each population aggregated across all strains. D20 indicates the number of total copies in the top 20 clones as a fraction of total copies across all clones. Data show mean with each dot representing a single mouse; $n = 8$ mice. **b** Plot showing the level of overlap between clones in different sequencing libraries prepared from the same sample aggregated across all strains. For each comparison, each clone was only counted once (no weighting for clone size) and functional overlap was computed using the Jaccard Index. Data show mean with each dot representing a single mouse; $n = 8$ mice. **c** Plot showing somatic hypermutation (SHM) as a fraction of total clones for each subset aggregated across mouse strains. The numbers in parenthesis indicate the total number of clones in the given subset. Each clone counts once for each subset. If a clone overlaps in multiple subsets, the SHM for each subset was calculated just for the sequences in a given subset. **d** The SHM by clones is shown across each mouse strain/subset combination as in Fig. 3c. **e** Plot showing CDR3 length (in nucleotides) of clones aggregated across all strains. Each clone is counted once in each subset/strain combination; $n = 8$. **f** Heatmap showing usage of the 20 most frequent VH genes. Each clone is counted only once. Data are normalized by row and visualized in Morpheus using the default settings. **g** Plot showing clones (rows) that overlap between at least two B cell subsets (columns). Numbers along the right side of the plot indicate clone counts. Data from a single analysis of subsets from a Yaa-DKO mouse. **h** Venn diagram showing clonal overlap where numbers indicate clone counts in the different subset interactions. Data from a single analysis of subsets from a Yaa-DKO mouse. **i** Heatmap showing Jaccard similarity (fraction of clones that overlap between different two-subset comparisons). Diagonal values are excluded for scaling. Data from a single analysis of subsets from a Yaa-DKO mouse. **j, k** Female ZTCE-DKO mice were treated with tamoxifen to mark Tbet-expressing cells with tdTomato expression for 3d. Representative FACS plots (**j**) and quantifications (**k**) of zsGreen and tdTomato expression from the indicated populations. Data show mean ± SEM; $n = 5$ over 2 independent experiments; $p$-value by paired two-tailed $t$-tests.

4A–C). In line with the known upregulation of both Type I and Type II IFN signatures in SLE patients, female ABCs were enriched for IFNα and IFNγ responses and upregulated several IFN stimulated genes (ISGs) expressed in SLE PBMCs (Fig. 4b; Supplementary Fig. 4B)[37]. Male ABCs were instead enriched for pathways related to Rho GTPase signaling and platelet activation (Fig. 4c; Supplementary Fig. 4D, E). Given the profound effects of *Tlr7* duplication on the ABCs of DKO males, we next sorted ABCs from Yaa-DKO males and compared their transcriptome to that of ABCs from DKO males (Fig. 4d). Similar to what was observed in female ABCs, the top pathways upregulated in ABCs from Yaa-DKO males were those related to IFN responses (Fig. 4e; Supplementary Fig. 4F, G). ABCs from DKO males were instead enriched for genesets related to hemostasis and platelet activation (Supplementary Fig. 4H, I). Thus, enrichment for an ISG signature differentiates ABCs from DKO females and males and *Tlr7* duplication promotes the upregulation of ISGs in male ABCs.

We next employed ATAC-seq to investigate the chromatin landscape of ABCs derived from the different DKOs. We identified at least 85,000 peaks in ABCs from female, male, and Yaa-DKO male mice (Supplementary Fig. 4J; Supplementary Table 1; Supplementary Data 1–2). *Tlr7* overexpression induced sufficient changes in the chromatin landscape to enable a motif analysis of the differentially accessible regions (DAR) of Yaa-DKO male ABCs versus male ABCs. Peaks upregulated in ABCs from Yaa-DKO males were enriched for motifs known to be bound by IRFs and NFκB family members (Fig. 4f). Peaks upregulated in ABCs from DKO males were instead enriched for ETS binding sites (Supplementary Fig. 4K). Consistent with the transcriptional profiles and the enrichment in IRF binding motifs, loci that were differentially accessible in ABCs from Yaa-DKO males as compared to DKO males included a number of ISGs like *Cxcl11*, *Ifi44*, and *Ifitm3* (Supplementary Fig. 4L). Thus, differences in ISG expression by ABCs are accompanied by a differential enrichment for IRF binding motifs.

We hypothesized that the aberrant gain of an IFN signature by ABCs in females was linked to intrinsic alterations in the epigenetic signatures of these cells and was not simply due to exposure to an IFN-rich environment. To address this hypothesis, we employed the CUT&RUN technique to compare the global loss of a repressive chromatin mark, H3K27me3, at key regulatory loci in ABC and FoB populations sorted from the same mice. Principal component analysis (PCA) of H3K27me3 peaks in ABCs and FOBs from DKO females and males resulted in distinct

clustering of samples based on sex and B cell subsets (Supplementary Fig. 4M). Genome-wide comparison of ABCs and FOBs from DKO females showed distinct loss of H3K27me3 signals at several loci among which *Tbx21*, the classical ABC marker, showed strongest loss in F-ABCs (Fig. 4g; Supplementary Fig. 4N–P). ABCs from DKO males demonstrated a similar loss of H3K27me3 signals at *Tbx21* when compared to FOBs (Fig. 4g). Notably, however, only ABCs from females but not those from males showed selective loss of H3K27me3 marks at ISGs like the CXCL cluster (Fig. 4h; Supplementary Data 3) suggesting that ABCs in DKO females have intrinsic alterations in their epigenetic signatures that can predispose them to aberrantly gain an IFN signature.

Given that loss of H3K27me3 signals has been shown to accompany XCI, we next specifically investigated the H3K27me3 peaks on the X-chromosome of ABCs and FOBs from DKO females. To gain insights into distinct regulation at various loci on the X-chromosome, we plotted Wald's z-statistic (X-chromosome deactivation score) obtained from DEseq2 analysis on the 3092 H3K27me3 peaks on the X-chromosomes between ABCs and FoBs from DKO females. H3K27me3 X-chromosome deactivation score plot showed distinct loss of H3K27me3 at over 2092 loci in ABCs as compared to FoBs that also included regulatory loci surrounding the *Tlr7* gene (Fig. 4i). A closer look at the *Tlr7* locus showed that there was a total of 3 peaks (chrX:167770493-167776995, chrX:167827673-167830728 and chrX:168121697-168125788) within the *Tlr7* neighborhood that were deactivated (Fig. 4i). The deactivated loci were located far away from the *Tlr7* gene body but remained in the same topologically associated domains (TADs) suggesting selective deactivation of repressive H3K27me3 marks within the intra-TAD boundaries of the *Tlr7* gene locus. These results suggest that selective loss of repressive chromatin marks in ABCs of DKO females can contribute to the increased expression of X-chromosome linked genes like *Tlr7*.

**Aberrant expansion of CD11c-expressing pre-GC B cells and PBs in DKO females.** In addition to ABCs, populations of CD11c−expressing GC B cells and PB/PCs also accumulate to a greater extent in DKO females than DKO males suggesting that they also contribute to the sex-bias in disease development. Since the phenotypic and molecular characteristics of these populations are largely unknown, we investigated them in greater detail. CD11c$^+$ GC B cells shared several phenotypic features with ABCs including high expression of T-bet, Fcrl5, and Cxcr3 (Supplementary Fig. 5A). As compared to CD11c$^−$

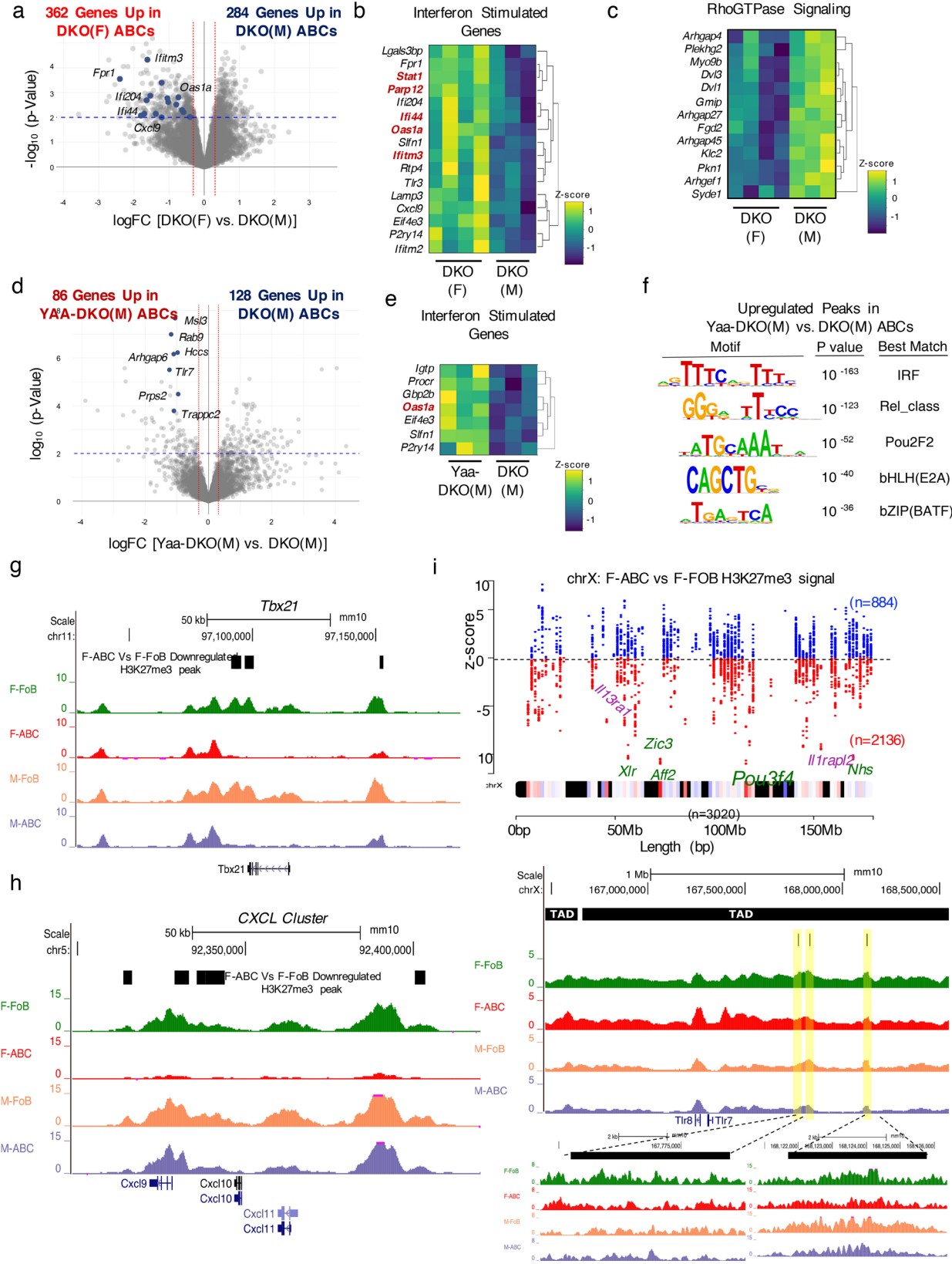

GC B cells, moreover, the transcriptome of sorted CD11c⁺ GC B cells was enriched for ABC genesets and for IFN responses (Fig. 5a, b; Supplementary Fig. 5B–C). Despite expressing comparable transcript levels of several classical GC target genes, including *Bcl6, Irf8*, and *Spib*, the levels of BCL6 protein were lower in the CD11c⁺ than in the CD11c⁻ populations (Fig. 5c;

Supplementary Fig. 5D). In line with the notion that intermediate expression of BCL6 protein in B cells has been associated with a pre-GC B cell state[38], the CD11c⁺ population contained increased frequencies of BCL6midIRF4⁺ B cells, a profile associated with pre-GC B cells, and was enriched in a pre-GC B cell signature by GSEA (Fig. 5d, e). BCR signaling, as

**Fig. 4 DKO ABCs upregulate ISGs in a sex-specific manner. a–e** RNA-seq analyses were performed on sorted ABCs (B220[+]CD19[+]CD11c[+]CD11b[+]) from aged (24+wk) female DKO (F), male DKO (M), and male Yaa-DKO (M) mice. **a** Volcano plot showing differentially expressed genes (p < 0.01 after Benhamini-Hochberg false discovery rate (FDR) was used to correct for multiple comparisons) in ABCs from DKO(F) and DKO(M) mice. Genes in blue show Interferon Stimulated Genes (ISGs) represented in Fig. 4b. **b** Heatmap showing the differential expression of ISGs in ABCs from DKO(F) and DKO(M) mice. **c** Heatmap showing differentially expressed genes in the REACTOME_RhoGTPase_CYCLE geneset. **d** Volcano plot showing differentially expressed genes (p < 0.01 after Benhamini–Hochberg false discovery rate (FDR) was used to correct for multiple comparisons) in ABCs from Yaa-DKO(M) mice and DKO(M) mice. Genes in blue show genes on the Yaa translocation. **e** Heatmap showing the differential expression of ISGs in ABCs from Yaa-DKO(M) and DKO(M) mice. **f** ATAC-seq was performed on ABCs from aged (24+wk) DKO(F), DKO(M), and Yaa-DKO(M) mice. Motif enrichment analysis in ATAC-seq peaks that are significantly upregulated (logFC >1.5; EdgeR, QLF FDR <0.05) in ABCs from Yaa-DKO(M) as compared to DKO(M) mice. **g–i** CUT&RUN for H3K27me3 marks was performed on ABCs from aged (24+wk) DKO(F) and DKO(M) mice. **g, h** Representative UCSC genome browser tracks of H3K27me3 marks for *Tbx21* gene (**g**) and *CXCL* gene cluster (**h**; *Cxcl9, Cxcl10 & Cxcl11*). The black boxes in each of the browser track represents genome co-ordinates of significantly downregulated H3K27me3 peaks from DKO(F) ABCs as compared to DKO(F) FoBs (2-fold down and = <0.05 FDR). **i** X-chromosome map of 3092 H3K27me3 peaks across the length the chromosome where each dot represents a H3K27me3 peak. Black bands on X-chromosome are devoid of H3K27me3 signal and the color spectrum represents various H3K27me3 peaks. The z-score (X-chromosome inactivation score) is a DESeq2 Wald statistic showing gain or loss of H3K27me3 signal across 3092 peaks. A total of 2135 peaks had negative z-scores (downregulated) in DKO(F) ABCs as compared to DKO(F) FoBs and 884 had positive z-scores (upregulated). Genome browser track of H3K27me3 signals over the *Tlr7* locus, highlighted in yellow are the downregulated H3K27me3 peaks in DKO(F) ABCs. The black boxes on the top represent TAD boundaries on the X-chromosome called by machine-learning program Peakachu[88] obtained from a murine B cell lymphoma CH12[89].

monitored by the phosphorylation of SYK and LYN, was furthermore significantly higher in CD11c[+] than CD11c[−] GL7[+] Fas[+] B cells (Fig. 5f). The CD11c[+] subset furthermore was less proliferative than CD11c[−] GL7[+]Fas[+] B cells (Fig. 5g, h). In contrast, the CD11c[+] population upregulated pathways related to migration and apoptotic cell clearance and expressed high levels of MerTK, a critical efferocytic receptor (Fig. 5i, j). A greater percentage of CD11c[+] GL7[+]Fas[+] B cells than CD11c[−] GL7[+]Fas[+] B cells furthermore could engulf apoptotic thymocytes and this was coupled with upregulation of surface MHC-II expression (Supplementary Fig. 5E, F). Thus, the CD11c[+] GL7[+]Fas[+] B cells that aberrantly expand in DKO females likely represent ABCs that have acquired a pre-GC B cell phenotype and can both engulf and present apoptotic debris, thus potentially augmenting autoreactive responses.

To gain insights into the molecular profiles of the CD11c[+] PB/PCs that also aberrantly expand in DKO females, we sorted CD11c[+] and CD11c[−] PB/PCs from DKO females and compared their transcriptome by RNA-seq (Fig. 6a). Consistent with their expression of CD11c, CD11c[+] PB/PCs were enriched for ABC signatures and IFN responses (Fig. 6b; Supplementary Fig. 6A–C). CD11c[+] PB/PCs furthermore expressed higher surface levels of ABC markers like Cxcr3 and Fcrl5 than CD11c[−] PB/PCs, although, consistent with the fate mapping studies, they expressed only low levels of T-bet (Fig. 6c). Thus, CD11c[+] PB/PCs share several transcriptional and phenotypic similarities with ABCs despite downregulating T-bet expression. CD11c[+] and CD11c[−] PB/PCs expressed similar transcript levels of *Prmd1* and *Irf4* although the transcriptional programs normally regulated by Blimp1 and Irf4 in PB/PCs were enriched to a greater extent in CD11c[−] PB/PCs than in CD11c[+] PB/PCs (Fig. 6d; Supplementary Fig. 6D). CD11c[+] PB/PCs were more proliferative than CD11c[−] PB/PCs and expressed higher levels of B220, MHC-II, and *Ciita* (Fig. 6e, f). As compared to CD11c[−] PB/PCs, CD11c[+] PB/PCs upregulated pathways related to migration including chemokine receptors like *Ccr3* as well as the expression of pro-inflammatory cytokines such as *Tnf* and *Il1b* and of inflammasome components like *Nlrp3* (Fig. 6h–j). Taken together these data thus suggest that CD11c[+] PB/PCs represent a population of PBs with distinctive migratory and pro-inflammatory characteristics.

**IRF8 and IRF5 cooperate in promoting the generation and differentiation of ABCs in DKO females.** The finding that a substantial proportion of effector B cell populations were related to ABCs and previously expressed T-bet prompted us to examine the role of T-bet in the accumulation of ABCs and their progeny in DKO females. We thus generated CD23-Cre *Tbx21*[flox/flox] DKO mice to specifically delete T-bet in B cells from DKO mice. Despite successful T-bet deletion, lack of B-cell T-bet did not significantly decrease the formation of ABCs as assessed by staining with CD11c and CD11b and other ABC markers such as Fcrl5 and Cxcr3 (Fig. 7a–c). Lack of B cell T-bet also did not affect the accumulation of total or CD11c[+] GL7[+]Fas[+] B cells and PB/PCs or the T$_{FH}$/T$_{FR}$ ratio (Fig. 7d–h). Lack of B cell T-bet did, however, result in a profound decrease in anti-dsDNA IgG2c without a corresponding increase in anti-dsDNA IgG1 (Fig. 7i). Thus, in autoimmune-prone DKO females, B cell T-bet is not necessary for ABC generation or differentiation but is specifically required for the production of IgG2c autoantibodies.

Given that global IRF5 deletion profoundly decreased ABC formation in DKO females[20,39], we next employed a similar strategy to investigate whether B-cell expression of IRF5 was specifically required for ABC generation and differentiation. Since IRF8 was also identified as a potential upstream regulator of ABCs in DKO females versus males (Supplementary Fig. 7A), we also extended this analysis to B cell IRF8. ABC accumulation was significantly decreased in CD23-Cre.Irf5[flox/+]DKO, CD23-Cre.Irf5[flox/flox]DKO, and CD23-Cre.Irf8[flox/flox]DKO females (Fig. 8a, b). Lack of B-cell IRF5 and IRF8 also markedly affected the expansion of GC B cells and CD11c[+] PB/PCs but only the absence of IRF5 affected the accumulation of total PB/PCs (Fig. 8c–f). Absence of B cell IRF5 or IRF8 also significantly impacted T$_{FH}$ responses and the production of anti-dsDNA IgG2c antibodies (Fig. 8g; Supplementary Fig. 7B, C). To delineate the relative contributions of IRF5 and IRF8 to the generation of ABCs, we utilized an in vitro culture system[20]. Lack of either IRF5 or IRF8 reduced the ability of DKO B cells to differentiate into ABCs albeit not in an identical manner, since deletion of IRF8 impaired CD11c upregulation while lack of IRF5 diminished T-bet induction (Fig. 8h, i). Both IRF5 and IRF8 were required for the enhanced production of CXCL10 by DKO ABCs and ChIP-qPCR demonstrated increased binding of both IRF5, as we previously reported[20], and IRF8 to the regulatory regions controlling the *Cxcl* cluster of genes in DKO B cells (Fig. 8j, k). Deleting IRF8 in DKO B cells impaired the ability of IRF5 to bind to these regions and lack of IRF5 decreased binding of IRF8 to these sites (Fig. 8k). Taken together these data support the notion that cooperation between IRF5 and IRF8 promotes the aberrant accumulation, function, and differentiation of ABCs in DKO females.

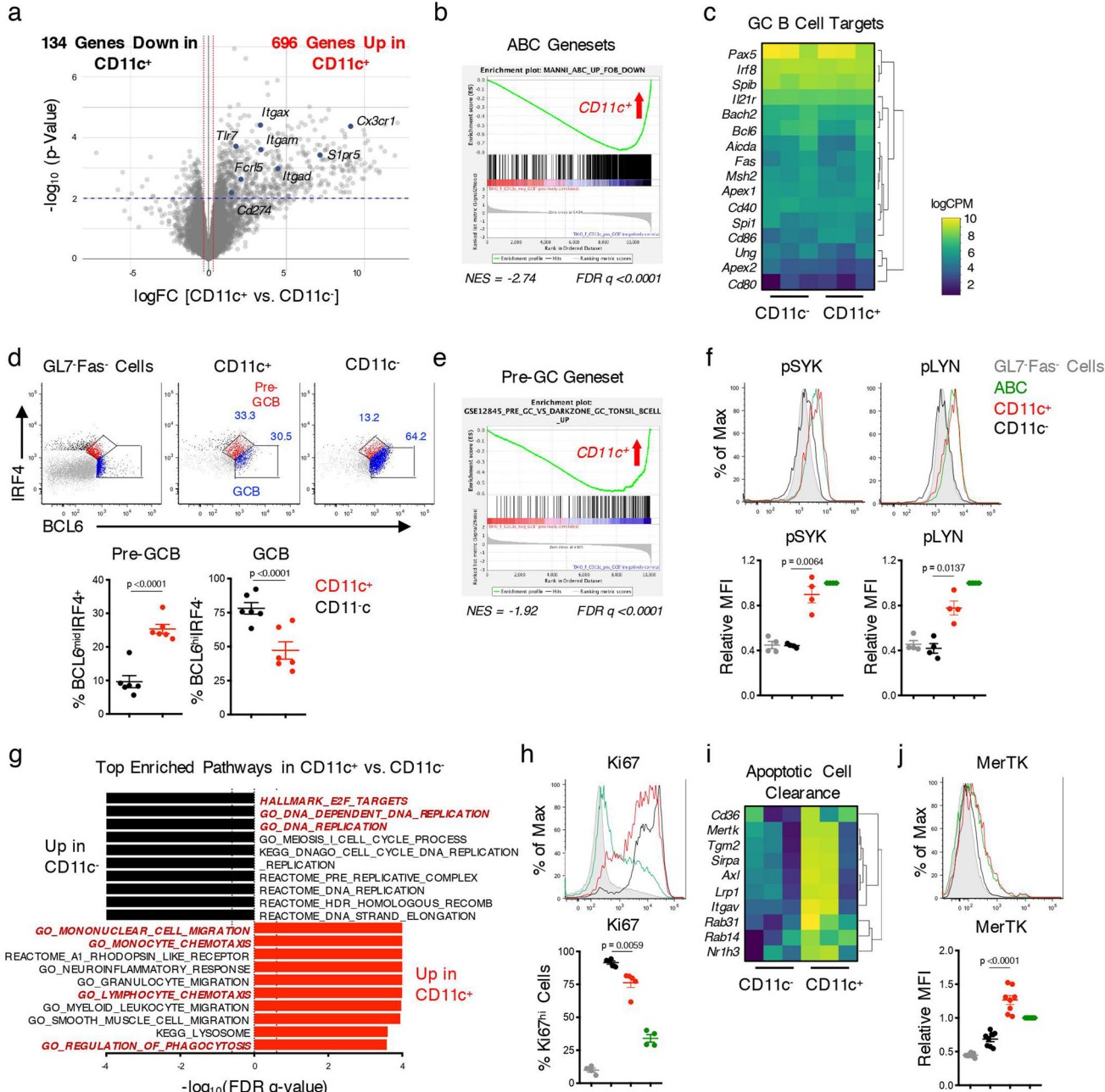

**Fig. 5 CD11c+ GC B cells exhibit a pre-GC B cell phenotype.** RNA-seq was performed on sorted CD11c+ and CD11c− GL7+CD38lo B cells (gated on B220+ splenocytes) from aged (24+wk) female DKO (F) mice. **a** Volcano plot showing genes differentially expressed (*p* < 0.01 after Benhamini−Hochberg false discovery rate (FDR) was used to correct for multiple comparisons) between CD11c+ B220+GL7+CD38lo (CD11c+) and CD11c− B220+GL7+CD38lo (CD11c−) cells. **b** Plot showing the enrichment of the ABC geneset from DKO mice in CD11c+ B220+GL7+CD38lo cells. **c** Heatmap showing the expression of GC B cell target genes in CD11c+ and CD11c− B220+GL7+CD38lo cells. **d** Representative plots and quantifications of pre-GC B cells (BCL6midIRF4+) and GC B cells (BCL6hiIRF4−) among CD11c+ CD19+GL7+Fas+ (CD11c+; red) and CD11c− CD19+GL7+Fas+ (CD11c−; black) cells from aged (24+wk) DKO(F) mice. Data show mean ± SEM; *n* = 6 from 2 independent experiments; *p*-value by paired two-tailed *t*-tests. **e** Plot showing the enrichment of a pre-GC geneset (GSE12845) in CD11c+ B220+GL7+CD38lo cells. **f** Representative histograms and quantifications of the phosphorylation of SYK(Y352) and LYN(Y416) in CD11c+ CD19+GL7+Fas+ cells (CD11c+; red), CD11c− CD19+GL7+Fas+ (CD11c−; black) cells from aged (24+wk) DKO(F) mice. CD19+CD11c+CD11b+ (ABCs; green) CD19+GL7−Fas− cells (gray) are shown as control. Data show mean ± SEM; *n* = 4 from 3 independent experiments; *p*-value by paired two-tailed *t*-test. **g** Plot showing the top pathways upregulated in CD11c− GL7+CD38lo cells (black) and CD11c+ GL7+CD38lo cells (red) by GSEA. Dotted line indicates significance threshold at FDR *q* < 0.25. **h** Representative histogram and quantification of Ki67 expression in the indicated populations from DKO(F) mice. Data show mean ± SEM; *n* = 5 from 3 independent experiments; *p*-value by paired two-tailed *t*-tests. **i** Heatmap showing the expression of genes related to apoptotic cell clearance in CD11c+ and CD11c− GL7+CD38lo cells from DKO(F) mice. **j** Representative histogram and quantification of MerTK expression on the indicated populations from aged (24+wk) DKO(F) mice. Data show mean ± SEM; *n* = 8 from 4 independent experiments; *p*-value by paired two-tailed *t*-tests.

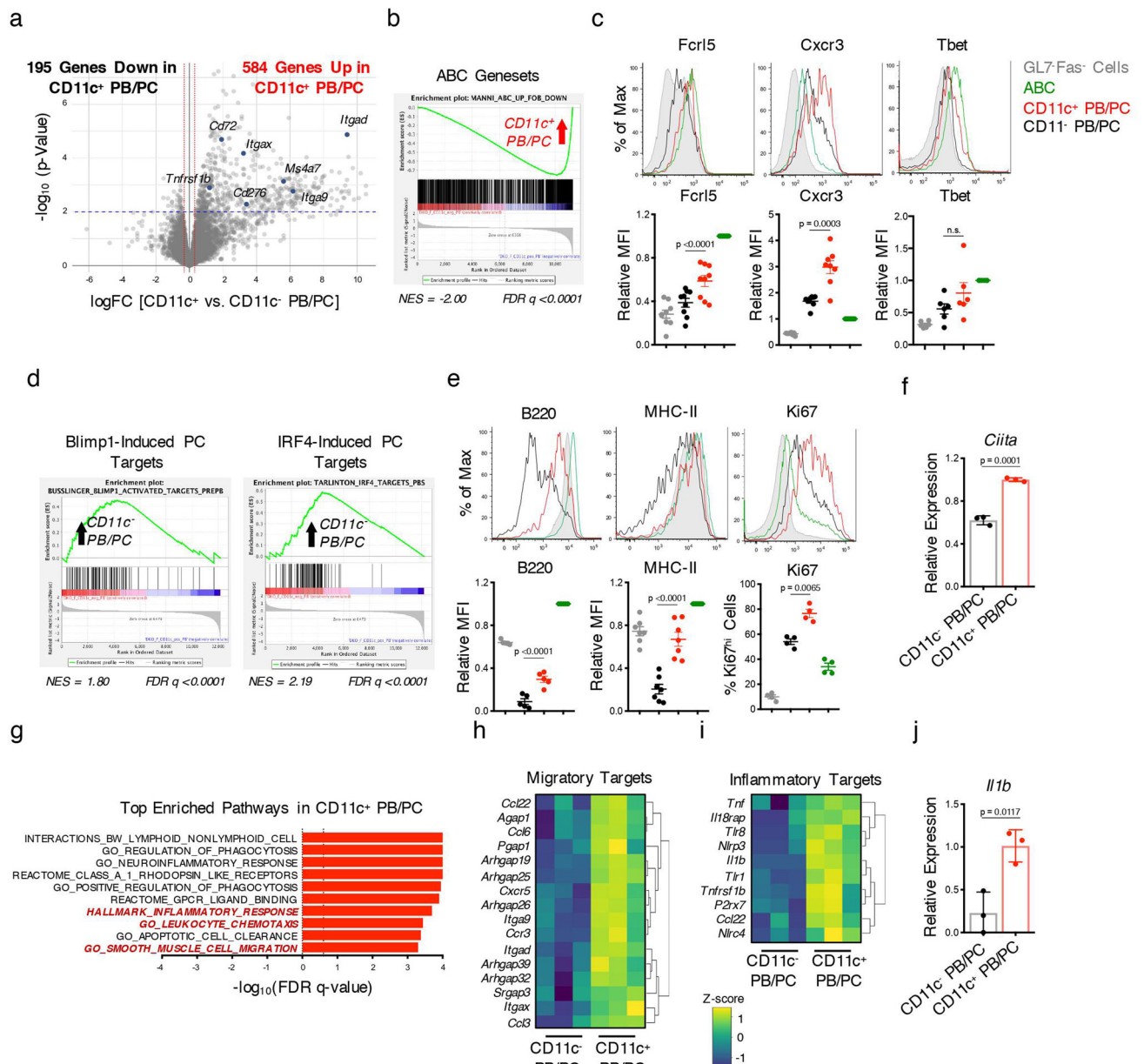

**Fig. 6 Expansion of CD11c+ PBs in DKO females. a** RNA-seq was performed on sorted CD11c+ and CD11c− PB/PCs (CD138+TACI+) from aged (24+wk) female DKO (F) mice. Volcano plot showing differentially expressed genes (p < 0.01 after Benhamini-Hochberg false discovery rate (FDR) was used to correct for multiple comparisons) between CD11c+ and CD11c− PB/PCs. Blue dots indicate ABC target genes upregulated in CD11c+ PB/PCs. **b** Plot showing the enrichment of the ABC geneset from DKO mice in CD11c+ PB/PCs. **c** Representative histograms and quantifications of Fcrl5, Cxcr3, and Tbet in CD11c+ PB/PCs (CD19mid/loCD138+; *red*) and CD11c− PB/PCs (black) from aged (24+wk) DKO(F) mice. CD19+CD11c+CD11b+ (ABCs) (green) and CD19+GL7−Fas− cells (gray) are shown as controls. Data show mean ± SEM; n = 6 for Tbet, n = 8 for CXCR3, n = 9 for FcRL5 over 3–5 independent experiments; p-value by paired two-tailed t-tests. **d** Plots showing the enrichment of Blimp1-induced[74] and IRF4-induced target genes in CD11c− PB/PCs by GSEA. **e** Representative histograms and quantifications of B220, MHC-II, and Ki67 expression in the indicated populations from aged (24+wk) DKO(F) mice. Data show mean ± SEM; n = 4 for Ki67, n = 5 for B220, n = 7 for MHC-II over 2–3 independent experiments; p-value by paired two-tailed t-tests. **f** Representative RT-qPCR showing *Ciita* expression in CD11c+ or CD11c− PB/PCs sorted from aged (24+wk) DKO(F) mice as in Fig. 6a. Data representative of 3 independent experiments and show mean ± SD; n = 3; p-value by unpaired two-tailed t-test. **g** Plot showing the top pathways enriched in CD11c+ PB/PCs as compared to CD11c− PB/PCs by GSEA. Dotted line indicates significance threshold at FDR q < 0.25. **h, i** Heatmap showing the expression of migratory (**h**) and inflammatory target genes (**i**) that are differentially expressed (p < 0.01) between CD11c+ and CD11c− PB/PCs in DKO(F) mice from Fig. 6a. **j** Representative RT-qPCR showing *Il1b* expression in CD11c+ or CD11c− PB/PCs sorted from aged (24+wk) DKO(F) mice as in Fig. 6a. Data representative of 3 independent experiments and show mean ± SD; n = 3; p-value by unpaired two-tailed t-test.

**TLR7 duplication in DKO males promotes ABC dissemination and severe immunopathogenesis.** While SLE preferentially affects females, males with lupus often exhibit a more rapid and severe course. Similarly, the shorter survival of Yaa-DKO males than DKO females suggested a more severe immunopathogenesis.

Interestingly, the early mortality and renal damage of Yaa-DKO males was accompanied by greater frequencies of ABCs in the blood and kidneys (Fig. 9a–c; Supplementary Fig. 8A, B). Furthermore, a histopathological analysis revealed that Yaa-DKO males also exhibited prominent inflammatory infiltrates in the

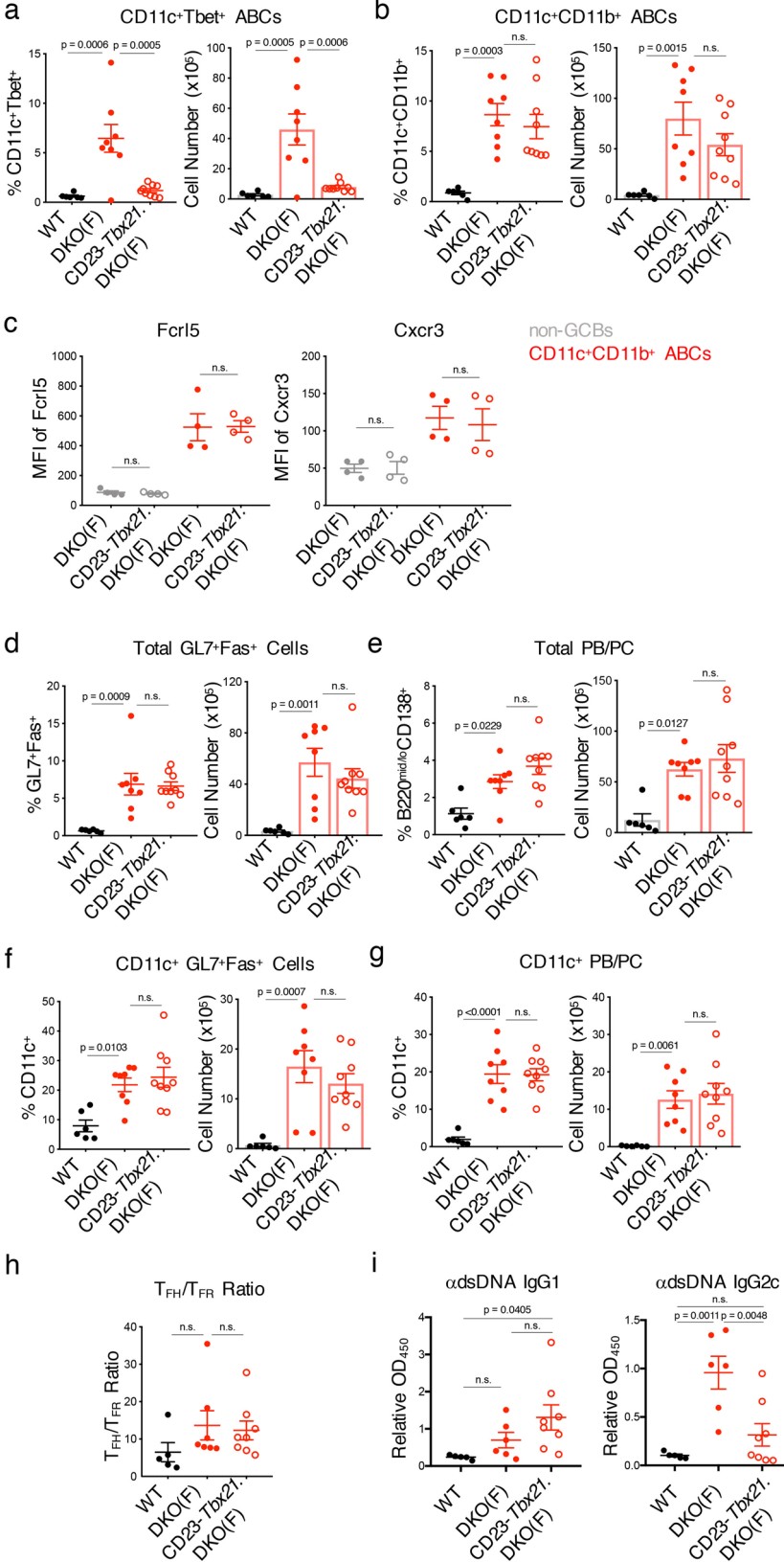

lungs although other organs, like the colon and the pancreas, were unaffected (Fig. 9d; Supplementary Fig. 8C). FACS analysis demonstrated that the lungs of Yaa-DKO males displayed a marked accumulation of ABCs, CD11c⁺ and CD11c⁻ GC B cells, and CD11c⁺ and CD11c⁻ PB/PCs as well as activated T cells (Fig. 9e; Supplementary Fig. 8D). DKO females also exhibited

lung inflammation, but as compared to age-matched Yaa-DKO males, the findings were less severe (Fig. 9d). Lung infiltrates in DKO females, however, worsened with age and were markedly ameliorated in aged CD23-Cre.*Irf8flox/flox*DKO females suggesting a crucial contribution of ABCs to the pulmonary inflammation (Fig. 9f). Thus, *Tlr7*-driven expansion of ABCs and their progeny

**Fig. 7 Tbet is not required for ABC accumulation in DKO mice. a, b** Quantifications showing the numbers of CD11c$^+$Tbet$^+$ ABCs (**a**) and CD11c$^+$CD11b$^+$ ABCs (**b**) from aged (24+wk) female C57BL/6 (WT; black circles), female DKO (F) (red circles), and female CD23-Cre.*Tbx21*f/f.DKO (CD23-*Tbx21*.DKO (F)) mice (red open circles). Data show mean ± SEM; $n = 6$ for WT, $n = 8$ for DKO(F), $n = 9$ for CD23-*Tbx21*.DKO(F) over 6 independent experiments; $p$-value by 1-way ANOVA followed by Tukey's test for multiple comparisons. **c** Plots showing the MFI of Fcrl5 and Cxcr3 on the surface of CD11c$^+$CD11b$^+$ ABCs (red) from the indicated mice. Non-GCBs (CD19$^+$GL7$^-$Fas$^-$; gray) are shown as control. Data show mean ± SEM; $n = 4$; $p$-value by unpaired two-tailed $t$-tests. **d–g** Quantifications showing the numbers of total GL7$^+$Fas$^+$ B cells (**d**), total PB/PCs (**e**), CD11c$^+$ GL7$^+$Fas$^+$ B cells (**f**), and CD11c$^+$ PB/PCs (**g**) from the indicated aged (24+wk) mice. Data show mean ± SEM; $n = 6$ for WT, $n = 8$ for DKO(F), $n = 9$ for CD23-*Tbet*.DKO(F) over 6 independent experiments; $p$-value by 1-way ANOVA followed by Tukey's test for multiple comparisons. **h** Quantifications showing the ratio of $T_{FH}$ to $T_{FR}$ cells ratio from the indicated mice. Data show mean ± SEM; $n = 5$ for WT, $n = 7$ for DKO(F), $n = 8$ for CD23-*Tbet*.DKO(F); $p$-value by 1-way ANOVA followed by Tukey's test for multiple comparisons. **i** Pooled ELISA data of anti-dsDNA IgG1 and IgG2c levels in the serum from the indicated aged (24+wk) mice. Data show mean ± SEM; $n = 5$ for WT, $n = 6$ for DKO(F), $n = 8$ for CD23-*Tbet*.DKO(F); $p$-value by 1-way ANOVA followed by Tukey's test for multiple comparisons.

can promote their accumulation in the lungs and the development of severe pulmonary inflammation.

Given that the marked lung inflammation in the setting of TLR7 dysregulation was reminiscent of the pathophysiology not only of SLE but also of severe viral infections such as COVID-19, we conducted additional hematologic and serologic analyses to assess whether other parameters known to be altered in this infection were similarly affected. A peripheral blood count demonstrated lower lymphocyte and platelet counts but increased monocytes in Yaa-DKO males than DKO males (Fig. 9g; Supplementary Fig. 8E). Yaa-DKO males also exhibited elevated levels of serum TNF and IL4 (Fig. 9h). We also assessed the production of antiphospholipid antibodies. While all DKOs produced higher levels of anti-phosphatidylserine (pS) IgM antibodies than WT controls, only Yaa-DKO males produced anti-pS IgG, anti-cardiolipin, and anti-MDA-LDL IgG (Fig. 9i). In line with the known ability of ABCs to produce antiphospholipid antibodies in response to pathogens and mediate hematologic abnormalities[40], anti-pS IgG and anti-cardiolipin antibodies correlated with ABC and PB/PC frequencies and there was a significant inverse association between ABC frequencies and platelet counts (Supplementary Fig. 8F–G). Thus, *Tlr7* duplication in DKO males results in hematologic and serologic abnormalities that can be associated not only with SLE but also with severe viral infections like COVID-19.

To gain insights into the molecular features that might result in a more rapid and severe TLR7-induced immunopathogenesis in Yaa-DKO males than in DKO females, we compared the transcriptomes of ABCs and their progeny sorted from Yaa-DKO males with those of the corresponding populations sorted from DKO females. Only minimal differences were observed by GSEA between the ABCs of DKO females and those of Yaa-DKO males (Supplementary Fig. 8H). However, a comparison of CD11c$^+$ pre-GC B cells and CD11c$^+$ PBs demonstrated that, as compared to DKO females, the populations derived from Yaa-DKO males were enriched for pathways involved in the production of antimicrobial peptides, cytokine interactions and signaling, and transcriptional regulation of granulopoiesis (Fig. 9j, k; Supplementary Fig. 8I–J). Thus, CD11c$^+$ effectors in Yaa-DKO males upregulate pathways that can enhance their pathogenicity and potentially enable their trans-differentiation.

## Discussion
The mechanisms that underlie the sexual dimorphism observed in responses to infections and vaccinations, and in autoimmune diseases remain incompletely understood. Here, we delineate sex-specific differences in the function and differentiation of ABCs, a subset of B cells that are emerging as critical mediators of antiviral responses and pathogenic players in autoimmunity. Using a spontaneous model of lupus where disease preferentially develops in females, we demonstrate that female ABCs exhibit a greater ability than male ABCs to accumulate, acquire an ISG signature,

and further differentiate into effector populations, which include CD11c$^+$ pre-GC B cells and CD11c$^+$ PBs. BCR sequencing and fate mapping reveal oligoclonal expansion and relatedness amongst ABCs, GC B cells, and PB/PC populations irrespective of the expression of CD11c. Genetic studies demonstrate a critical role for TLR7, IRF5, and IRF8 in promoting these abnormalities in females. Duplication of *Tlr7* in males overrides the sex-bias and triggers severe immunopathology marked by intense lung inflammation and early mortality. Thus, sex-specific differences permeate several aspects of ABC biology in autoimmune settings suggesting that this compartment may be uniquely endowed to function in a sex-specific manner.

Our studies demonstrate marked differences in the ability of female and male DKO ABCs to express an ISG signature, which has recently been shown to be upregulated in ABCs and PCs from SLE patients and is one of the best-known features not only of this disease but also of antiviral responses[7,41–43]. The ISG signature observed in SLE patients indeed overlaps with that detected upon viral infections and immunizations and an inability to upregulate ISGs has recently been shown to distinguish severe from mild-to-moderate COVID-19 patients[44–46]. Our epigenetic and genetic analyses furthermore indicate that this system is critically reliant on the dysregulation of IRF5 and IRF8 activity, well-known controllers of ISGs whose variants have long been associated with lupus pathogenesis and whose activity can be a target of viral evasion strategies[46–48]. While we cannot rule out that TLR7-driven IFNα production by pDCs could contribute to the differential expression of the ISG signature in DKO females and males, loss of repressive epigenetic markers at ISGs were selectively observed in ABCs, but not FoBs, from DKO females supporting the notion that acquisition of this signature is controlled in large part by cell-intrinsic mechanisms rather than exposure to an IFN-rich environment.

Sex-based differences were also observed in the ability of ABCs to further differentiate into effector subsets. Despite similar $T_{FH}$ responses to DKO males, DKO females exhibited a more robust expansion of GC B cells and PB/PCs, which in addition to classical CD11c$^-$ subsets, also included CD11c$^+$ expressing subsets. BCR sequencing and fate mapping uncovered surprising relationships of ABCs not only with CD11c$^+$ pre-GC B cells and CD11c$^+$ PBs, but also with CD11c$^-$ GC B and CD11c$^-$ PC, which exhibited remarkably different transcriptional profiles from the CD11c$^+$ B cell populations. The IRFs may again be a crucial component of this dysregulation. Indeed, the known ability of IRFs to homo/heterodimerize and target ISREs as well as interact with other transactivators like the Ets protein PU.1[47] may be well-suited to help confer a high degree of heterogeneity to ABCs and their progeny and fine-tune their transcriptional profiles. The pairing of IRF5 with IRF8 in this compartment may provide ABCs and their CD11c$^+$ progeny with a more "innate" quality than traditional CD11c$^-$ B cell subsets and facilitate their distinctive combination of innate

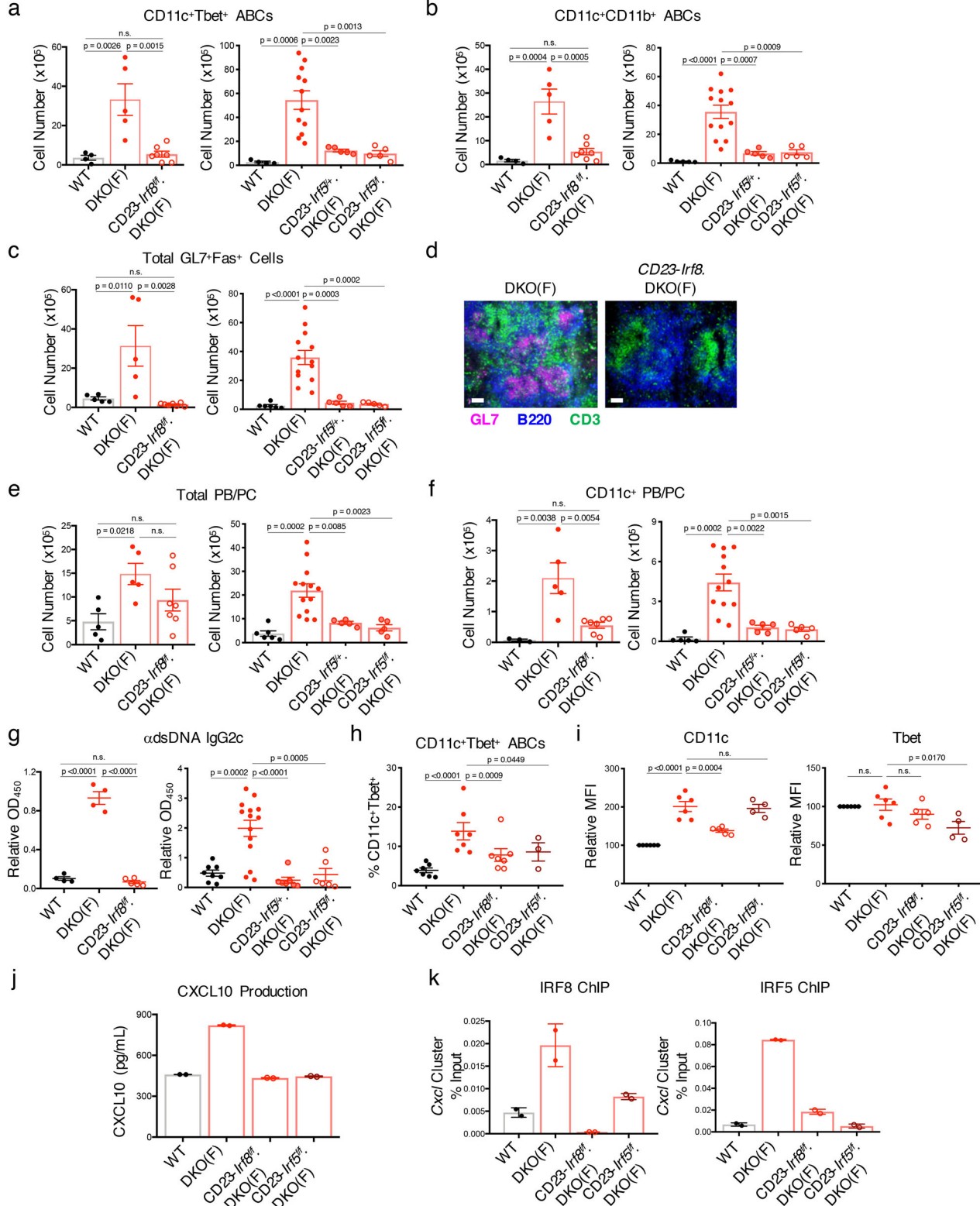

and adaptive functions. Changes in IRF8 activity, like those mediated by ROCK2 phosphorylation[49], could instead result in the downregulation of typical ABC transcriptional targets like CD11c and help promote its interaction with other transactivators enabling ABCs to give rise to both CD11c$^+$ and CD11c$^-$ effector progeny. In support of this notion preliminary studies indicate that TLR7 engagement can inhibit ROCK2 activation thus directly affecting this balance.

In contrast to the marked effects observed upon deletion of IRF5 and IRF8, lack of B-cell T-bet in DKOs exerted more selective effects suggesting that, in this autoimmune setting, the generation and differentiation of CD11c$^+$ B cells is less reliant on T-bet. T-bet however played a critical role in the acquisition of functional capabilities, such as the production of anti-dsDNA IgG2a/c antibodies, that have been associated with these cells. Given that a small fraction of FoBs was found to express T-bet in

**Fig. 8 IRF8 and IRF5 are required for ABC differentiation in DKO females. a–c** Quantifications showing the numbers of CD11c+Tbet+ ABCs (**a**), CD11c+CD11b+ ABCs (**b**), and total GL7+Fas+ B cells (**c**) from the spleens of aged (24+wk) female C57BL/6 (WT), female DKO(F), female CD23-Irf8.DKO(F), female CD23-Irf5f/+.DKO(F), and female CD23-Irf5f/f.DKO(F) mice. Data show mean ± SEM; n = 4 for WT, n = 5 for DKO(F), CD23-Irf5f/+.DKO(F), and CD23-Irf5.DKO(F), n = 7 for CD23-Irf8.DKO(F), n = 12 for DKO(F) over 6–7 independent experiments; p-value by 1-way ANOVA followed by Tukey's test for multiple comparisons. **d** Representative immunofluorescence images of B220 (blue), GL7 (pink), and CD3 (green) on the indicated mice. Data representative of 4 mice per genotype. Bars show 50 μm. **e, f** Quantifications showing the numbers of total PB/PCs (**e**; B220mid/loCD138+) and CD11c+ PB/PCs (**f**) from the indicated aged (24+wk) mice. Data show mean ± SEM; n = 5/6 for WT, n = 5/13 for DKO(F), n = 7 for CD23-Irf8.DKO(F), n = 5 for CD23-Irf5f/+ (**e**); n = 3/5 for WT, n = 5/12 for DKO(F), n = 7 for CD23-Irf8.DKO(F), n = 5 for CD23-Irf5f/+ (**f**) over 6–7 independent experiments; p-value by 1-way ANOVA followed by Tukey's test for multiple comparisons. **g** ELISA data showing anti-dsDNA IgG2c in the serum from the indicated aged (24+wk) mice. Data show mean ± SEM; n = 4/8 for WT, n = 4/14 for DKO(F), n = 5 CD23-Irf8.DKO(F), n = 7 for CD23-Irf5+/-.DKO(F), n = 6 for CD23-Irf5f/f.DKO(F); p-value by 1-way ANOVA followed by Tukey's test for multiple comparisons. **h–k** CD23+ B cells from the indicated young (8–12wk) mice were stimulated for 2–3d with αIgM, αCD40, and IL-21. **h** Quantification of CD11c+Tbet+ B cells after 3d culture. Data show mean ± SEM; n = 7 for WT, DKO(F), and CD23-Irf8.DKO(F), n = 3 for CD23-Irf5.DKO(F); p-value by 1-way ANOVA followed by Tukey's test for multiple comparisons. **i** Quantifications showing the expression of CD11c and Tbet in the indicated strains after 3d culture as in Fig. 4h. Data show mean ± SEM; n = 6 for WT and DKO(F), n = 5 for CD23-Irf8.DKO(F), n = 4 for CD23-Irf5.DKO(F); p-value by 1-way ANOVA followed by Tukey's test for multiple comparisons. **j** ELISA data showing CXCL10 production in the supernatants after 3d culture as in Fig. 4h. Data representative of 3 independent experiments and show mean ± SD. **k** Representative ChIP-qPCR of IRF8 and IRF5 binding at the Cxcl cluster in the indicated young (8–12wk) mice. Data representative of 3 independent experiments and show mean ± SD.

the fate mapping studies, additional analyses will furthermore be required to establish whether these T-bet+ cells are destined to acquire a full-fledged ABC phenotype or whether they represent a separate pool of T-bet+ B cells that can also differentiate into a heterogenous progeny and are critical for autoantibody production.

VH sequencing demonstrated profound oligoclonal expansion and further confirmed common clonal relationships between ABCs, GC B cells, and PB/PCs, which again could be observed within both CD11c+ and CD11c− compartments. Interestingly, different degrees of clonal overlap could be observed between ABCs and different progenies within distinct mice suggesting that while both extrafollicular and GC-like differentiation pathways may be available to these cells, the precise routes employed by ABCs to undergo terminal differentiation can vary depending on the specific inflammatory milieus that they are exposed to. Although the spontaneous GCs observed in this autoimmune setting displayed some atypical features, as evidenced by the finding that these GCs were exquisitely sensitive to the absence of IRF8 unlike those observed upon T-dependent immunization[50], the ABC-derived GC populations did exhibit higher levels of SHM than ABCs suggesting that these GCs were functional. Notably, several of the VH regions overexpressed in DKO ABCs and their progeny, including VH1 (J558) and VH14 (SM7), have previously been associated with the production of lupus autoAbs and with the expanded ABCs of SLC−/− mice, another spontaneous autoimmune model[51].

The plasticity of ABCs and a limited capability to acquire GC-like as well as PB phenotypes have been previously observed in Ehrlichia and influenza infection models[9,10,52] suggesting that, upon encountering a pathogen, these differentiation pathways are available to ABCs, albeit in a restricted manner. One of the consequences of TLR7 overexpression in our autoimmune model furthermore was the dissemination of ABCs in the blood and their accumulation in the lungs, a pattern that was also transiently exhibited by T-bet+ B cells shortly after influenza infection[9]. Strikingly, the lung infiltrates in aged DKO females were markedly ameliorated in mice lacking IRF8 in B cells supporting a key role for ABCs and their progeny in promoting the pulmonary inflammation observed in DKOs. Thus, some of the pathogenesis of SLE may reflect a breakdown in the regulatory mechanisms aimed at restricting the differentiative routes and dissemination of ABCs in time and space. In this regard, the ability of the SWEF proteins to coordinate cytoskeletal organization and IRF function[21] may be particularly important. Indeed, by ensuring the proper positioning, cell-cell interactions, and transient migration

of ABCs and by restricting the accessibility to IRF controlled transcriptional programs, these molecules could help ensure that a rapid initial response is coupled with the transient generation of a pool of progeny, which is diverse but limited in size.

Although lower levels of TLR7 responsiveness by male ABCs may protect DKO males from the development of lupus, the transcriptional profile of male ABCs showed enrichment for Rho GTPase signaling pathways. Dysregulation of these pathways has been linked to well-known disorders such as hypertension and cardiovascular disease, which also show a sexually dimorphic pattern but preferentially affect males[53,54]. This raises the intriguing possibility that lower levels of TLR7 engagement by male ABCs could not only result in suboptimal antiviral defenses but also promote a distinct set of pathogenic responses marked by vascular inflammation and thrombosis. Surprisingly, TLR7 duplication in Yaa-DKO males resulted in more profound immunopathogenesis than that observed in DKO females suggesting that females may have evolved mechanisms to contain the overwhelming inflammatory effects that may accompany the greater levels of TLR7 stimulation that they are predisposed to due to incomplete XCI. In this regard, it is interesting to note that both our studies and recent work in humans suggest that ABCs may be uniquely susceptible to partially escape XCI[55]. We cannot, however, rule out that the differences between Yaa-DKO males and DKO females primarily result from complete rather than partial escape from XCI secondary to the translocation of the X-chromosome segment onto the Y-chromosome. While our studies have highlighted a crucial role for TLR7 in mediating the sex differences in the development of lupus in this model, sex hormones like estrogen and androgen have also been shown to regulate B cell responses including their positioning and survival and to modulate disease pathogenesis[56,57] and thus are also important contributors to these differences.

Intriguingly, several of our findings are reminiscent of the pathophysiology of COVID-19 where disease outcomes have shown striking age- and sex-dependent differences[1]. Indeed, expansion of T-bet+ B cells (including activated naïve and DN) and PBs, which in some cases can exhibit reprogramming potential and lower levels of SWAP-70, have all been observed in severe COVID-19 patients[12,13,58,59]. Some of the features known to accompany disease severity in COVID-19 such as lymphopenia, thrombocytopenia, production of antiphospholipid antibodies, and a broad array of cytokine responses[60,61] could be observed in Yaa-DKO males in the absence of any viral infections suggesting that these effects could directly result from unbridled TLR7 stimulation. Whether the reported ability of ABC-like cells

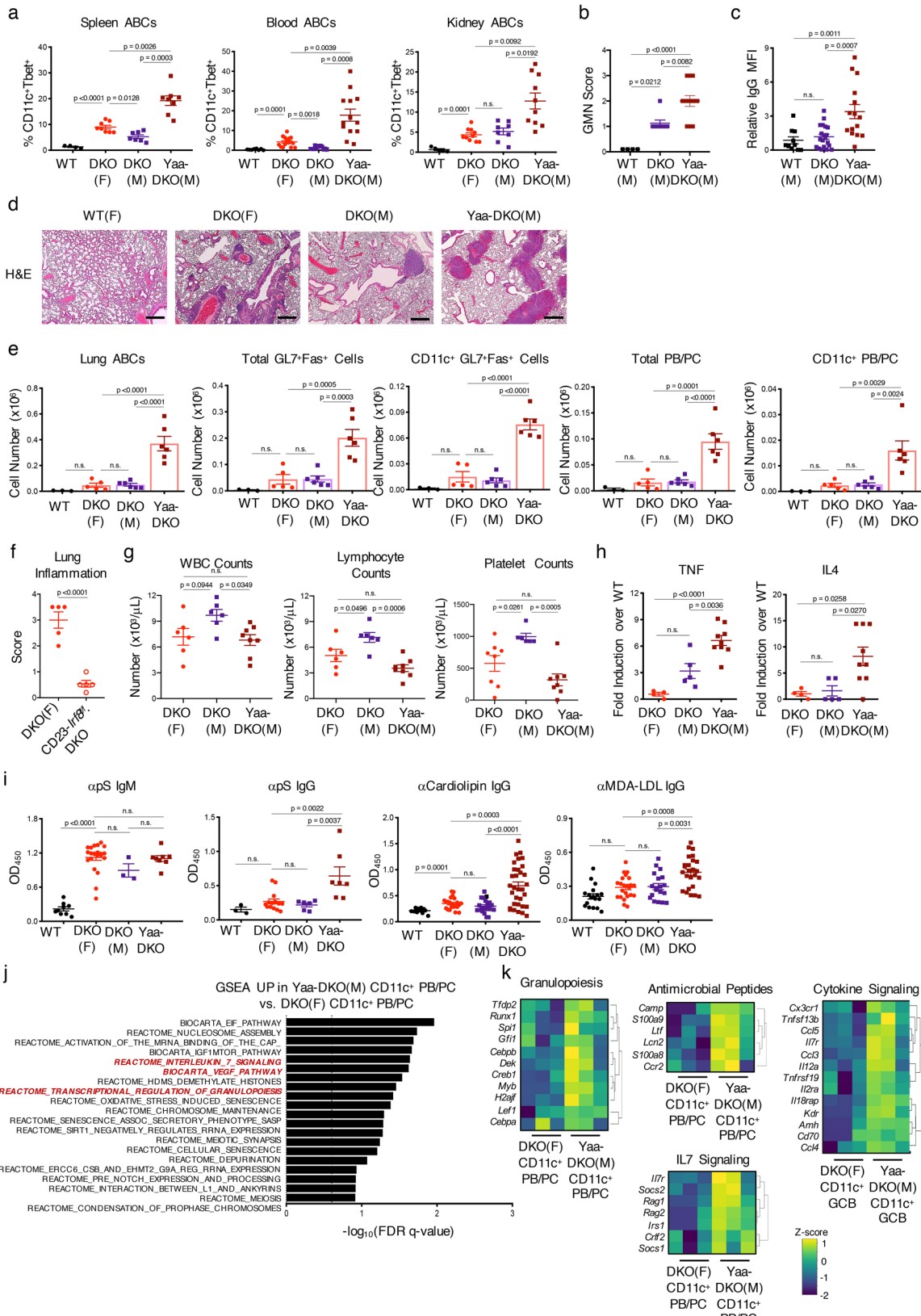

to accumulate with age in adipose tissue in an IL-1-dependent manner[62] could provide a ready depot of these cells and lower the threshold of TLR7 stimulation needed by these cells to acquire full pathogenic functions will be an important question to be addressed given the link between obesity and COVID-19 outcomes. The marked sex differences in the pathways utilized by female and male ABC subsets suggest that the effectiveness of approaches targeting these cells may demonstrate sex-dependent differences. Such a scenario may be encountered, for instance, in the case of statins or geranylgeranylation inhibitors that can inhibit Rho-GTPases and have vaccine adjuvant properties[63]. Sex-specific differences in ABC function and differentiation could

**Fig. 9 *Tlr7* duplication promotes ABC dissemination and immunopathogenesis. a** Quantification of CD11c⁺Tbet⁺ ABCs from the spleens, blood, and kidneys of aged (24+wk) female C57BL/6 (WT; *black circles*), female DKO(F) (*red circles*), male DKO(M) (*purple squares*), and male Yaa-DKO(M) (*maroon squares*) mice. Data shows mean ± SEM; *n* = 4/8/5 for WT, *n* = 8/15/10 for DKO(F), *n* = 7/12/8 for DKO(M), *n* = 8/13/10 for Yaa-DKO(M) over 5 independent experiments; *p*-value by 1-way Brown–Forsythe and Welch ANOVA followed by Games–Howell's test for multiple comparisons. **b** Glomerulonephritis (GMN) scores of kidneys from the indicated aged (24wk+) mice. Data shows mean ± SEM; *n* = 4 for WT, *n* = 8 for DKO(M), *n* = 12 for Yaa-DKO(M); *p*-value by 1-way ANOVA followed by Tukey's test for multiple comparisons. **c** Quantification of IgG deposition in the kidneys from the indicated aged mice. Data shows mean ± SEM; *n* = 10 fields across 2 mice for WT, *n* = 20 fields across 4 mice for DKO(M), *n* = 15 fields across 3 mice for Yaa-DKO(M); *p*-value by 1-way ANOVA followed by Tukey's test for multiple comparisons. **d** Representative H&E images of lungs from the indicated aged (30wk+) mice. Data representative of 2 WT, 4 DKO(F), 4 DKO(M), and 4 Yaa-DKO(M) mice. Bars show 50 μm. **e** Quantifications of the numbers of CD11c⁺CD11b⁺ ABCs, total GL7⁺Fas⁺ B cells, CD11c⁺ GL7⁺Fas⁺ B cells, total PB/PCs, and CD11c⁺ PB/PCs in the lungs from the indicated aged (24wk+) mice. Data show mean ± SEM; *n* = 3 for WT, *n* = 5 for DKO(F), *n* = 6 for DKO(M) and Yaa-DKO(M) over 2 independent experiments; *p*-value by 1-way ANOVA followed by Tukey's test for multiple comparisons. **f** Plot showing the inflammation score in the lungs of 45wk-old female CD23-Cre.Irf8^f/f.DKO(F) and female Irf8^f/f.DKO(F) control mice as determined by H&E staining. Data show mean ± SEM; *n* = 5; *p*-value by unpaired two-tailed *t*-test. **g** Plots showing the numbers of white blood cells, lymphocytes, and platelets in the blood from the indicated aged (24wk+) mice. Data show mean ± SEM; *n* = 6 for DKO(F) and DKO(M), *n* = 8 for Yaa-DKO(M); *p*-value by 1-way ANOVA followed by Tukey's test for multiple comparisons. **h** Luminex data showing the fold-increase in serum levels of TNF and IL4 in aged (24wk+) DKO(F), DKO(M), and Yaa-DKO(M) mice relative to WT. Data show mean ± SEM; *n* = 4 for DKO(F), *n* = 5 for DKO(M), *n* = 9 for Yaa-DKO(M); *p*-value by 1-way ANOVA followed by Tukey's test for multiple comparisons. **i** ELISA data for anti-phosphatidylserine (pS) IgM and IgG, anti-Cardiolipin IgG, and anti-MDA-LDL IgG in the serum from the indicated aged (24wk+) mice. Data show mean ± SEM; for anti-pS IgM, *n* = 9 for WT, *n* = 22 for DKO(F), *n* = 3 for DKO(M), *n* = 8 for Yaa-DKO(M); for anti-pS IgG *n* = 3 for WT, *n* = 14 for DKO(F), *n* = 6 for DKO(M), *n* = 7 for Yaa-DKO(M); for anti-Cardiolipin IgG, *n* = 11 for WT, *n* = 23 for DKO(F), *n* = 20 for DKO(M), *n* = 32 for Yaa-DKO(M); for anti-MDA-LDL IgG, *n* = 18 for WT, *n* = 23 for DKO(F), *n* = 19 for DKO(M), *n* = 25 for Yaa-DKO(M); *p*-values by 1-way ANOVA followed by Tukey's test for multiple comparisons. **j** RNA-seq was performed on sorted CD11c⁺ PB/PCs (CD138⁺TACI⁺) from Yaa-DKO(M) mice as in Fig. 6a. Plot shows top pathways enriched in CD11c⁺ PB/PCs from Yaa-DKO(M) mice as compared to CD11c⁺ PB/PCs from DKO(F) mice as determined by GSEA. Dotted line indicates significance threshold at FDR q < 0.25. **k** Heatmaps showing the expression of genes enriching the granulopoiesis, antimicrobial peptide, cytokine signaling, and IL7 signaling genesets in Yaa-DKO CD11c⁺ PB/PCs.

thus not only contribute to the well-known sex-bias that underlies several autoimmune diseases but also broadly impact responses to pathogens and vaccination.

## Methods

**Mice.** All mice used in this study were on a C57BL/6 background. DEF6-deficient (*Def6*^tr/tr) mice were generated by Lexicon Pharmaceuticals, Inc. using a gene-trapping strategy. Integration of the gene-trapping construct occurred in the first intron of the *Def6* gene, downstream of the exon coding for the initiation methionine. Targeted ES cells (129SvEv) were injected into C57BL/6 blastocysts to generate chimeric mice, which were then backcrossed to C57BL/6 mice for >10 generations[22]. Swap70-deficient (*Swap70*^−/−) were generated via a targeting construct designed to replace a 2.7-kb SacI/SmaI fragment containing exon 1 of the SWAP-70 gene with a phosphoglycerate kinase (PGK)-neo resistance cassette[22]. *Def6*^tr/tr*Swap70*^−/− (DKO) mice were generated by crossing *Def6*^tr/tr mice with *Swap70*^−/− mice that had been backcrossed onto C57BL/6 background for >10 generations[22]. C57BL/6, *Irf8*^f/f mice (#014175), and *Tbx21*^f/f mice (#022741) were from Jackson Laboratory. B6.SB-*Yaa*/J were originally provided by Derry Roopenian and are available through Jackson (Yaa; #000483), *Tlr7*-deficient were originally provided by Eric Pamer and are available through Jackson (*Tlr7*^−/− #008380)[64]. Male Yaa mice were crossed with female DKO mice to generate Yaa-DKO male mice[65]. *Irf8*^f/f mice, *Tbx21*^f/f mice, and *Tlr7*^−/− mice were crossed with DKO mice to generate *Irf8*^f/f.DKO, *Tbx21*^f/f.DKO mice, and *Tlr7*^−/−.DKO mice. Generation of DKO Blimp1-YFP and *Irf5*^f/f DKO mice were previously described[20,66]. CD23-cre mice were provided by Jayanta Chaudhuri and were previously described[67]. Tbet-zsGreen-T2A-CreER^T2-Rosa26-loxP-STOP-loxP-tdTomato DKO (ZTCE-DKO) mice were generated by crossing DKO mice with Tbet-zsGreen-T2A-CreER^T2 provided from Jinfang Zhu[36] and with B6.Cg-*Gt (ROSA)26Sor*^tm14(CAG-tdTomato)Hze/J (#007914, Jackson). ZTCE-DKO mice were treated with Tamoxifen by oral gavage 3d before experiments were conducted. All mice used in the experiments were kept under specific pathogen-free conditions. Control animals for most experiments were bred separately, but co-housed. Experiments comparing DKO(M) mice with Yaa-DKO(M) mice were performed with animals that were not co-housed. Mice were euthanized using carbon dioxide. All animal experiments were approved by the Institutional Animal Care and Use Committee of the Hospital for Special Surgery and WCMC/MSKCC and the experiments were carried out following these established guidelines.

**Antibodies and flow cytometry.** The following monoclonal antibodies to mouse proteins were used for multi-parameter flow cytometry: B220-PB or B220-APC/Cy7 (RA3-6B2; 1:400), CD4-PerCP/Cy5.5 (RM4-5; 1:400), CD45-PerCP/Cy5.5 (HI30; 1:400), CD11b-PE/Cy7 or CD11b-FITC (M1/70; 1:400), CD11c-APC/Cy7 or CD11c-APC (N418; 1:400), CD19-PB or CD19-PE (HIB19; 1:400), CD21-APC (7E9; 1:200), CD23-PE or CD23-PerCP/Cy5.5 (B3B4; 1:200), CD44-PE/Cy7 or CD44-A700 (IM7; 1:200), CXCR3-A488 (CXCR3-173; 1:200), MHC-II-PerCP/Cy5.5 (AF6-120.1; 1:600), and Tbet-PE or Tbet-PE/Cy7 (4B10; 1:800) were

obtained from BioLegend. Streptavidin-conjugated antibodies were also obtained from BioLegend. Antibodies to BCL6-PE or BCL6-PE/Cy7 (K112-91; 1:100), CD138-APC (281-2; 1:1200), CXCR5-Biotin (2G8; 1:200), Fas-Biotin (Jo2; 1:200), FcRL5-FITC (509F6; 1:200), and GL7-FITC (1:600) were obtained from BD. Antibodies to Foxp3-APC (FJK-16s; 1:100), IgD-FITC (11-26; 1:500), IgM-PE/Cy7 (II/41; 1:1000), IRF4-FITC (3E4; 1:200), IRF8-PE (V3GYWCH; 1:200), Ki67-PE (solA15; 1:800), MerTK-APC (DS5MMER; 1:200), and PD1-PB (J43; 1:200) were obtained from eBioscience. For intracellular staining, cells were fixed after surface staining at 4 °C with the Transcription Factor Staining Kit (eBioscience; #00-5523-00) following the manufacturer's instructions. For intracellular cytokine staining, splenocytes were stimulated with 50μg/mL PMA and 1μM Ionomycin for 4 hr. Cells were incubated with BrefeldinA for the final 3 h of stimulation. After stimulation, cells were fixed and permeabilized with a Transcription Factor Staining Kit (eBioscience; #00-5523-00) and stained using anti-IFNγ-APC (BioLegend; XMG1.2; 1:200) and recombinant mouse IL21R Fc Chimera (R&D; 1:600) followed by PE-labeled affinity-purified F(ab')₂ fragment of goat anti-human Fcγ (Jackson ImmunoResearch). For detection of phosphorylated antigens, splenocytes were fixed in BD Fixation Buffer (#554714) for 20 min at RT. Cells were then washed and permeabilized in 90% methanol for 30 min at −20 °C and then incubated with antibodies against phosphorylated SYK (Y352; Cell Signaling) or LYN (Y416; Cell Signaling) for 45 min at room temperature. LysoTracker staining was done before surface staining with 20 nM LysoTracker-Deep Red (Thermo; L12492) for 45 min at 37 °C in RPMI 1640. All data were acquired on a BD FACS Canto and analyzed with FlowJo (TreeStar) software.

**Cell sorting.** Single-cell suspensions were prepared from spleens of the indicated mice. For sorting ABCs, splenocytes were pre-enriched for B cells with B220 microbeads (Miltenyi Biotec; #130-049-501) following manufacturer's instructions. B cells were stained with CD11c-APC (N418; 1:400), CD11b-FITC (M1/70; 1:400), CD19-PB (HIB19; 1:400), B220-APC/Cy7 or B220-PE/Cy7s (RA3-6B2; 1:400), and CD23-PE (B3B4; 1:200) and were sorted on FACS Aria or Influx (BD). ABCs were collected for RNA or cultured for B cells with 1μM Imiquimod in RPMI 1640 medium (Corning) supplemented with 10X FBS (Atlanta Biologicals), 100 U/mL Penicillin (Corning), 100 mg/mL Streptomycin (Corning), 1X non-essential amino acids (Corning), 2mM L-Glutamine (Corning), 25 mM HEPES (pH7.2-7.6; Corning), and 50 μM β-Mercaptoethanol. For sorting CD11c⁺ and CD11c⁻ GC B-like cells and PB/PCs, splenocytes were pre-enriched using biotinylated antibodies against B220 and CD138 and with streptavidin-conjugated microbeads (Miltenyi Biotec; #130-048-101) following manufacturer's instructions. Cells were stained with CD11c-APC/Cy7 (N418; 1:400), CD19-PB (HIB19; 1:400), CD138-APC (281-2; 1:600), CD38-PE/Cy7 (90; 1:600), GL7-FITC (1:600), and TACI-PE (8F10; 1:400) and were sorted on FACS Aria or Influx (BD).

**B cell cultures.** CD23⁺ B cells were purified from single cell suspensions of splenocytes with biotinylated anti-CD23 (BD Bioscience; #553137) and streptavidin microbeads (Miltenyi Biotec; #130-048-101) according to manufacturer's instructions. Cells were cultured for 2–3 d in RPMI 1640 medium (Corning)

supplemented with non-essential amino acids (Corning), 2 mM L-Glutamine (Corning), 25 mM HEPES (pH 7.2–7.6), and 50 μM β-Mercaptoethanol and stimulated with 5 μg/mL F(ab')$_2$ anti-mouse IgM (Jackson ImmunoResearch), 5 μg/mL purified anti-mouse CD40 (BioXcell), and 50 ng/mL IL-21 (Peprotech). For thymocyte engulfment assays[68] thymocytes were harvested and isolated from 4–6 week-old WT mice and treated with 50 μM Dexamethasone for 4 h to induce apoptosis. Apoptotic thymocytes were then stained with 1 μM CypHer5E (GE Healthcare) for 45 min at 37 ºC in serum-free Hank's Balanced Sodium Solution (HBSS). Stained thymocytes were co-cultured with splenocytes at a 1:10 splenocyte to thymocyte ratio. Apoptotic thymocytes were removed by washing with cold PBS and splenocytes were assessed for efferocytosis by flow cytometry. MHC-II expression was compared on efferocytic (CypHer5E$^+$) and non-efferocytic (CypHer5E$^-$) B cells.

**Real-time RT-PCR and chromatin immunoprecipitation assays**. Total RNA was isolated using the RNeasy Plus Mini kit (Qiagen; #74134). cDNAs were prepared using the iScript cDNA synthesis kit (Biorad; #1708841). Real-time PCR was performed using the iTaq Universal SYBR Green Supermix (Biorad; #1725121). Gene expression was calculated using the ΔΔCt method and normalized to *Ppia*. For chromatin immunoprecipitation (ChIP) assays, B cell cultures were harvested at d2 and chromatin extracts were prepared using the truChIP Chromatin Shearing Reagent Kit (Covaris; #520154). 100μg of sonicated DNA protein complexes were used for immunoprecipitations with anti-IRF5 (Abcam #ab21689), anti-IRF8 (Cell Signaling; #5628), or normal anti-rabbit Ig control antibodies. DNA purified from the immunoprecipitates and inputs was analyzed by qPCR. All primers are listed in Supplementary Table 2.

**ELISAs and antigen microarrays**. For anti-dsDNA ELISA, plates were coated with 100 μg/mL salmon sperm DNA (Invitrogen; #AM9680) at 37 ºC overnight and blocked in 2% BSA in PBS at room temperature for 2 h. For anti-cardiolipin and anti-phosphatidylserine ELISA, Immulon 2HB plates (Thermo) were coated with 75μg/mL of cardiolipin or with 30 μg/mL phosphatidylserine dissolved in 100% ethanol overnight. Sera were diluted 1:200 and incubated on coated plates at 25 ºC for 2 h. Supernatants from sorted ABCs were used neat after 7 d culture with 1 μM Imiquimod. Plates were then incubated with HRP-labeled goat anti-mouse IgG or IgG2c Fc antibody for 1 h (eBioscience). OD$_{450}$ was measured on a microplate reader. Autoantibody activities against a panel of autoantigen specificities were measured using an Autoantigen Microarray platform developed by the University of Texas Southwestern Medical Center[69,70].

**Histology and immunofluorescence staining**. Tissue specimens were fixed in 10% neutral buffered formalin and embedded in paraffin. Tissue sections were stained with periodic acid-Schiff (PAS) or with hematoxylin and eosin (H&E) and analyzed by light microscopy. The nephritis scoring system was adapted from the International Society of Nephrology/Renal Pathology Society (ISN/RPS) classification of human lupus nephritis. The final score accounted for morphological pattern (mesangial, capillary, membranous) and for the percentage of involved glomeruli. For immunofluorescence staining, kidneys or spleens were embedded in OCT and frozen in 2-methylbutane surrounded by dry ice. Frozen blocks were cut into 9 μm section with cryotome and stored at −80 ºC. Upon thawing, sections were let dry at room temperature and stained. Spleen sections were stained with B220 (BD; RA3-6B2) and GL7 (BioLegend). Kidney sections were stained with FITC-labeled goat anti-mouse IgG (Jackson ImmunoResearch). Specimens were captured by Q capture software on a Nikon Eclipse microscope and quantifications were calculated using ImageJ software.

**RNA-sequencing**. Quality of all RNA and library preparations were evaluated with BioAnalyzer 2100 (Agilent). Sequencing libraries were sequenced by the Epigenomics Core Facility at Weill Cornell Medicine using a HiSeq2500, 50-bp single-end reads at a depth of ~15–50 million reads per sample. Read quality was assessed and adapters trimmed using FASTP[71]. Reads were then mapped to the mouse genome (mm10) and reads in exons were counted against Gencode v27 using STAR2.6 Aligner[72]. Differential gene expression analysis was performed in R using edgeR3.24.3. Genes with low expression levels (<2 counts per million in at least one group) were filtered from all downstream analyses. Differential expression was estimated using quasi-likelihood framework. Benhamini-Hochberg false discovery rate (FDR) procedure was used to correct for multiple testing. Genes with an unadjusted *p*-value less than 0.01 were considered differentially expressed. Downstream analyses were performed in R using a visualization platform built with Shiny developed by bioinformaticians at the David Z. Rosensweig Genomics Research Center at HSS.

Gene set enrichment analysis was performed using GSEA software (Broad Institute)[73]. Genes were ranked by the difference of log-transformed count per million (cpm) for contrasted conditions. Molecular Signatures DataBase v7.0 (Broad Institute) was used as a source of gene sets with defined functional relevance. Gene sets were also curated from RNA-seq datasets of ABCs from female DKO mice and from the blood of SLE patients[15,20] and from previously published PB/PC datasets[74,75]. Gene sets ranging between 15 and 2500 genes were included into the analysis. Nominal *p*-values were FDR corrected and gene sets with FDR

*q* < 0.25 were used to crease GSEA enrichment plots. Analyses of differentially expressed genes were also performed using the online webtool CPDB and upstream regulator analyses were conducted using the Enrichr databases[76–78].

**ATAC-sequencing**. The nuclei of sorted ABCs from female DKO, male DKO, or Yaa-DKO mice were prepared by incubation of cells with nuclear preparation buffer (0.30 M sucrose, 10 mM Tris pH 7.5, 60 mM KCl, 15 mM NaCl, 5 mM MgCl$_2$, 0.1 mM EGTA, 0.1% NP40, 0.15 mM spermine, 0.5 mM spermidine, and 2 mM 6AA)[74]. For library preparation[79], fragments were amplified using 1× NEB-next PCR master mix and 1.25 μM of custom Nextera PCR primers with the following PCR conditions: (1) 72 ºC for 5 min (2) 98 ºC for 30 s and (3) 3 cycles of 98 ºC for 10 s, 63 ºC for 30 s, and 72 ºC for 1 min. To reduce GC and size bias, the PCR reaction was monitored using qPCR in order to stop amplification before saturation. Additional amplification was performed for a total of 10–12 cycles. The libraries were purified using a Qiagen PCR cleanup kit yielding a final library concentration of ~30 nM in 20 μL. Paired-end 50 bp sequences were generated from samples with an Illumina HiSeq2500 and, following adapter trimming with FastP, were aligned against mouse genome (mm10) using bowtie2 with –local -q -p options. Peaks were called with MACS2 with—*macs2 callpeak -f BAMPE—nomodel —shit -100—extsize 200—B—SPMR -g $GENOMESIZE -q 0.01* options. Peak-associated genes were defined based on the closest genes to these genomic regions using RefSeq co-ordinates of genes. We used the *annotatePeaks* command from HOMER to calculate ATAC-seq tag densities from different experiments and to create heatmaps of tag densities. Sequencing data were visualized by preparing custom tracks for the UCSC Genome browser. De novo transcription factor motif analysis was performed with motif finder program *findMotifsGenome* from HOMER package on ATAC-seq peaks. Peak sequences were compared to random genomic fragments of the same size and normalized G+C content to identify motifs enriched in the targeted sequences.

**CUT&RUN analysis**. Freshly sorted ABCs and FoBs from both male and female DKO mice were used for the preparation of CUT&RUN libraries as described in[80] with slight modifications. We used three biological replicates for each condition. Sorted cells were immediately incubated overnight with H3K27me3 (anti- Tri-Methyl-Histone H3 (Lys27) (Cell Signaling Technology #Rabbit mAb #9733) or IgG isotype controls (Guinea Pig anti-Rabbit IgG (H+L) Secondary Antibody (novusbio #NBP1-72763) at 1:50 dilution of antibody binding buffer with a final concentration of 0.05% digitonin. Following incubation and digestion CUTANA™ pAG-MNase from EpiCypher (SKU: 15-1116) the digested DNA fragments were release and purified by QIAquick PCR purification kit (#2810). The resulting DNA fragments were end-repaired and barcoded libraries were made with the NEB Ultra II DNA Library Prep kit ((E7645S, E7335S & E7500S)). The libraries were sequenced on Illumina NextSeq 500 to a minimum of 15 million reads per sample. The data was analyzed using CUT&RUNTools pipeline[81] and resulting peak calls from SECAR were used for subsequent analysis. The peaks from IgG isotype controls were excluded in the SECAR peak calling program. A master file of H3K27me3 peaks across all the conditions and replicated was created and ncbi/BAMscale[82] was used to quantify peaks and generate scaled coverage tracks for viewing in UCSC genome browser. The raw counts on each H3K27me3 peak were obtained using ncbi/BAMscale cov –-bed master.H3K27me3.bed –prefix Peak_-count –bam *.bam command and DESeq2 was employed to obtain differential H3K27me3 peaks between F-ABC and F-FOB mice. computeMatrix and plotHeap map functions were used from deepTools package to plot average tag densities and heatmap. Tag densities for the peaks in individual samples was calculated from DESeq2 and plotted as heatmap using Morpheus heatmap tool. The volcano plots were plotted with EnhancedVolcano[83] available on Bioconductor and X chromosome heat of H3K27me3 signals was plotted using chromoMap. The z-statistic used in X chromosome H3K27me3 inactivation score heat map was obtained by implementing DESeq2 Wald test comparing F-ABC over F-FoB by taking log2 fold-change and dividing by its standard error resulting in z-statistic and a p-value is computed reporting that the probability that a z-statistic is small enough to reject null hypothesis. We summarized the differential peaks obtained by DESeq2 comparisons between F-ABC, F-FoB, M-ABC and M-FoB in Supplementary Data 3.

**IgH sequencing and immune repertoire analysis**. Genomic DNA was extracted from sorted cells using the QIAGEN Gentra DNA purification kit (Qiagen, #158689). Primer sequences and library preparation were described previously[9]. Samples were amplified in duplicate (2 biological replicates per sample) using 100 ng of input DNA per replicate (Supplementary Data 4). Sequencing was performed on an Illumina MiSeq instrument in the Human Immunology Core facility at the University of Pennsylvania using a 2x300bp paired-end kit. Sequencing data analysis was done as previously described[9]. Sequences were quality controlled with pRESTO[84]. Briefly, paired reads were aligned, sequences with low quality scores were discarded, and base calls with low confidence were masked with "N"s. IgBLAST was then used to align sequences to V- and J-genes in the IMGT database[85]. To group related sequences together into clones, ImmuneDB hierarchically clusters sequences with the same VH gene, same JH gene, same CDR3 length, and 85% identity at the amino acid level within the CDR3 sequence[86]. Clones with consensus CDR3 sequences within 2

nucleotides of each other were further collapsed to account for incorrect gene calls. Sequencing data were submitted to SRA under project number PRJNA663307 in accordance with the MiAIRR standard[87]. Prism v8.4.3 was used for D20, Jaccard index, SHM, CDR3 length, and VH gene usage histogram plots. Morpheus (Broad Institute) was used for VH heatmap. Venn Diagrams were generated in (http://www.interactivenn.net/). Other calculations were performed as described previously[86].

**Statistics**. All plots show datapoints from independent mice pooled across multiple experiments, unless otherwise noted. p-values were calculated with two-tailed t-tests or ANOVA followed by multi-group comparisons, as indicated in the figure legends. Survival data was tested by Kaplan–Meyer analysis with significance determined by the log-rank (Mantel–Cox) test. Correlation data was tested by paired Pearson correlations. p-values of <0.05 were considered significant. Ns: not significant, $*p < 0.05$, $**p < 0.01$, $***p < 0.001$, $****p < 0.0001$. Statistical analysis was performed with Graphpad Prism 8.

**Reporting summary**. Further information on research design is available in the Nature Research Reporting Summary linked to this article.

## Data availability

Sequence data that support the findings of this study have been deposited in NCBI's Gene Expression Omnibus (GEO) with the primary accession code GSE175365 and to SRA under project number PRJNA663307. Other reagents and materials are available upon reasonable request from the corresponding author. Source data are provided with this paper.

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

## Acknowledgements

We thank members of the HSS Research Institute for thoughtful discussions and reagents. This work was supported by the US National Institutes of Health (AR064883 and AR070146 to ABP; T32 Rheumatology Research Training Grant to E.R.; P30 CA016520 and P30 AI0450080 to E.L.P.), the Rheumatology Research Foundation, the Lupus Research Alliance, the Peter Jay Sharp Foundation, the Tow Foundation which provided support for the David Z. Rosensweig Genomics Research Center, Giammaria Giuliani and the Ambrose Monell Foundation, the Barbara Volcker Center for the Michael D. Lockshin Fellowship (M.M.), and Marina Kellen French and the Anna-Maria and Stephen Kellen Foundation (D.J. and P.S.). Technical support was provided by the Epigenomics Core, the Microscopy and Imaging Core, and the Flow Cytometry Core Facility of Weill Cornell Medicine, by the Laboratory of Comparative Pathology at Memorial Sloan Kettering Cancer Center, by the Human Immunology Core at the Perelman School of Medicine, and by the Genomics and Microarray Core Facility at the University of Texas Southwestern Medical Center, and from the Office of the Director of the National Institutes of Health under Award Number S10OD019986 to Hospital for Special Surgery.

## Author contributions

E.R. designed and performed the experiments, interpreted the experiments, and wrote the manuscript; M.M. designed and performed the experiments and interpreted the experiments; D.F.C., D.J., S.G., and J.R.C. performed the experiments; W.M., A.R., and E.L.P. conducted the IgH sequencing bioinformatics analyses; T.P. assisted with the histological analyses; M.B. performed the CUT&RUN bioinformatics analyses; Y.C. performed the RNA-seq and ATAC-seq bioinformatics analyses; P.K.S. helped write the manuscript; R.J. generated the $Swap70^{-/-}$ mice and helped write the manuscript; A.B.P. designed and supervised the study, interpreted the experiments, and wrote the manuscript.

## Competing interests

The authors declare no competing interests.
