## [Peer Review File · Nature Communications]

REVIEWER COMMENTS

Reviewer #1 (Remarks to the Author):

The authors build on an interesting line of research on the role of ABCs in autoimmunity by providing interesting information on the sexual dimorphism of systemic autoimmunity. The study has been performed elegantly and presented in a clear manner.

Reference to SARS-Cov2 in the abstract does not make sense because there is no clear evidence for sexual dimorphism in COVID-19. There are no data on SARS-Cov2.

The same in the introduction. It can be discussed in the discussion section but the sexual dimorphism issue is not clear, at least the way I understand it.

The discussion is too long. Yet, since TLR7 is one of the factors contributing to sexual dimorphism in lupus, can the authors discuss its relative value and mention other contributors which have been entertained in the literature?

Reviewer #2 (Remarks to the Author):

ABCs are a unique subset of B cells defined by surface CD11b and CD11c expression and the transcription factor Tbet. They are expanded in the context of age and autoimmune disease, and it has been suggested that an analogous B cell population exists in humans (DN2: CD27-CD21lo) and is associated with SLE as well as some viral infections. ABCs require TLR7 for their development (as well as Ag, T cells) and represent an antigen-experienced population that harbors SHM. They rapidly produce IgG2b/c Abs in response to stimulation with TLR agonists or Ifng or Il-21. For all these reasons it has been suggested that they are a unique "memory" B cell population.

Pernis and colleagues previously described a lupus-like disease with expansion of ABCs in DKO mice lacking the SWEF proteins, Def6 and SWAP-70 (Nat Imm 2018). Def6 and SWAP-70 are Rho GEFs that also localize with TFs in the nucleus and have a previously defined role in regulating IRF4 function. The 2018 report defines the chromatin landscape and transcriptome of WT and DKO ABCs and show (via genetic epistasis, RNAseq, and ATACseq) that the SWEF proteins normally repress expansion of ABCs by negatively regulating IRF5. Challenges included isolating effects of germline and conditional B cell deletion strategies to cell-intrinsic roles within ABCs.

In the current manuscript, the authors take a similar set of approaches (genetic epistasis and extensive genome-wide profiling of ABCs and other B cell populations). They begin by observing that autoimmune disease and ABC expansion in this DKO model has a female sex-bias reminiscent of human SLE. They show that TLR7 is required for both ABC expansion and disease in DKO mice, and that adding an extra copy of TLR7 via Yaa overcomes the sex bias in DKO males. They show that disease is dependent on both IRF5 and IRF8 (but only in part on Tbet). Finally, they profile BCRs, transcriptome and epigenome of ABCs and other effector B cell subsets from these animal models. They show that sex bias and TLR7-dose correlates with an ISG signature in ABCs, and they also identify clonal relatedness between ABCs and effector compartments like GC and PC.

Overall - the role of TLR7 dose in mouse models of SLE has been well-established - dating back to study of Yaa-FcgRIIb disease by the Bolland lab (Pisitkun et al. Science 2007). The requirement of TLR7 for generation of ABCs is also previously shown, and as noted above, contribution of IRF5 to DKO ABC expansion and disease was already established in prior Nat Imm 2018 paper (albeit with CD21cre rather than CD23cre). Finally, the contribution of TLR7 signaling via IRFs to IFN production in context of viral infection and SLE models is also described in the literature. However, the strength of the current report is the mechanistic connection across all of these pathways in the context of an extremely well-characterized and profoundly sex-biased SLE model. The report moreover identifies heterogeneity within effector B cell populations at the level of CD11c expression and associated transcriptomes, and identifies extensive clonal relatedness between ABCs and effector compartments like GC and PC. Importance of dissecting these pathways in the DKO model rests on identification of ABCs as a relevant cell population with pathogenic potential across models and in human disease, as well as genetic associations between IRF5, Def6, ifn

pathway, and SLE. For all of these reasons, the current report is of broad interest to researchers in the field of SLE.

The manuscript frames a very extensive and impressive body of work that leans equally on genetic epistasis and transcriptome analyses to develop a mechanistic model for sex-biased disease pathogenesis in DKO animals. The conclusions are supported by the data overall. Queries and comments below are enumerated in detail, but a key question to highlight is the extent to which TLR7 dose is acting in a B cell-intrinsic or ABC-intrinsic manner in this model. Presumably it is to some extent, but there may be a contribution by other type 1 ifn-producing cell populations like pDC that is not addressed. Another question that is hard to address with existing tools is extent to which all machinery studied genetically is operating specifically within ABCs, and extent to which ABCs are indeed the primary pathogenic cell population in this model. While circumstantial evidence supporting role of ABCs is substantial (and increased by the work presented here), these caveats ought to be discussed.

Specific critiques and/or queries:

-Introduction notes "...has recently been shown to escape X-chromosome inactivation" about TLR7. Ref (4) Sci Immunology report from 2018 describes biallelic expression of TLR7 in about 20-30% of several immune cell populations which is striking, but this doesn't imply 2-fold expression of TLR7 broadly. Therefore, I would modify phrase to "... escape complete X-chromosome inactivation" for those readers that aren't familiar with the primary report.

Related to this, it is worth highlighting that presumably Yaa male DKO mice have most severe disease than female DKO mice because x-inactivation is altogether absent with X-segment transposed to Y.

-S1a assesses TLR7 transcript in male and female Fo B cells and ABCs. If incomplete x-inactivation is posited to account for higher expression in ABCs, why is this not the case in Fo B cells?

Figures 1,2. Evidence for TLR7-dose accounting for sex-specific differences in DKO model is compelling and use of TLR7^{-/-} v. Yaa was an elegant choice. While effects of TLR7 are likely ABC-intrinsic at least in part, this isn't formally demonstrated and since impaired x-inactivation of TLR7 is also reported (cited ref 4) in pDCs, excessive Ifn production by that cell type represents another possible contributing mechanism. Related to this: In Figure 4, the enrichment for ISG signature among female and Yaa ABCs from DKO mice makes sense and could represent TLR7 dose affecting type 1 ifn production by pDCs. It should be noted that impact of TLR7/sex may therefore not be entirely cell-intrinsic per se. Indeed, ISG signature might be enhanced among multiple cell types within these animals.

Figure 2. Massive spontaneous PC expansion in Yaa DKO males is striking. What isotype does this represent; what do total Ig levels look like? Is this PB/PC expansion exclusively in spl or also in BM (since latter harbors LLPCs, would have different implications about origin of the PCs)?

Figure 3. Attempt to lineage track formerly Tbet⁺ ABCs with "ZTCE" construct is a very nice complement to clonal relatedness studies. However, looks like Fo B cells can express Tbet too so caveat here is that the origin among ABCs is not definitive. Compelling experiments, but would include this caveat or address it.

Figure 7. Uncoupling ABC accumulation and autoantibody production by B cell-specific Tbet deletion would either argue that ABCs depend on other transcriptional regulators in DKO mice OR it could suggest that ABCs are not the origin and do not drive autoantibody production. Both possibilities should be discussed. For this reason, the Tbet result seems important to present in main figure rather than only in the supplement. This also raises the interesting point of what Tbet expression contributes to the ABC transcriptome. This seems a more significant question than the DEG between CD11c⁺ and CD11c⁻ PB/PCs from Fig 6.

Reviewer #3 (Remarks to the Author):

This is an extensive body of work revealing and then mechanistically understanding the role of

TLR7 in sex-specific bias observed in ABCs derived from DEF6 and SWAP70 double knockout mice. The immunologic aspects of this work are solid in my opinion, but I have the following suggestions pertaining the epigenetic work performed.

1) It is understood that Tlr7 can escape X-chromosome inactivation in B cells, but it is not clear if and why this would be augmented in DEF6/SWAP70 DKO mice. It will be of interest to show if there is a difference in X-inactivation in the Tlr7 locus between DKO and wild type mice. This can be done using bisulfite sequencing in the Tlr7 locus where regulatory regions in mice have already been established. In addition, average DNA methylation levels in this locus comparing male and female DKO mice will indirectly reveal if there is evidence of incomplete X-inactivation in female mice specifically in ABCs and not FoBs in keeping with the Tlr7 mRNA expression results provided.

2) Additional details should be provided regarding the ATAC-seq experiments. For example, how many million reads were generated per sample? How was tagmentation quality assessed?

3) Using ATAC-seq data, the authors can directly assess transcription factor binding differences, using an analysis such as HINT-ATAC to perform TF footprint analysis if enough reads were generated per sample. This will provide a more direct evidence for the TF arguments made using the enrichment analysis in accessibility peaks.

Reviewer #1 (Remarks to the Author):

The authors build on an interesting line of research on the role of ABCs in autoimmunity by providing interesting information on the sexual dimorphism of systemic autoimmunity. The study has been performed elegantly and presented in a clear manner.

Reference to SARS-Cov2 in the abstract does not make sense because there is no clear evidence for sexual dimorphism in COVID-19. There are no data on SARS-Cov2. The same in the introduction. It can be discussed in the discussion section but the sexual dimorphism issue is not clear, at least the way I understand it. The discussion is too long. Yet, since TLR7 is one of the factors contributing to sexual dimorphism in lupus, can the authors discuss its relative value and mention other contributors which have been entertained in the literature?

We thank the Reviewer for the positive comments on our manuscript and truly appreciate the suggestions offered. In response to the Reviewer's suggestions, we have removed the specific reference to SARS-CoV2 in the abstract and introduction and only broadly mentioned the impact of sex-differences on antiviral responses. We have also mentioned other key contributors, such as sex-hormones, to the sex-bias of SLE while trimming the discussion by removing the last paragraph.

Reviewer #2 (Remarks to the Author):

ABCs are a unique subset of B cells defined by surface CD11b and CD11c expression and the transcription factor Tbet. They are expanded in the context of age and autoimmune disease, and it has been suggested that an analogous B cell population exists in humans (DN2: CD27-CD21lo) and is associated with SLE as well as some viral infections. ABCs require TLR7 for their development (as well as Ag, T cells) and represent an antigen-experienced population that harbors SHM. They rapidly produce IgG2b/c Abs in response to stimulation with TLR agonists or Ifng or Il-21. For all these reasons it has been suggested that they are a unique "memory" B cell population.

Pernis and colleagues previously described a lupus-like disease with expansion of ABCs in DKO mice lacking the SWEF proteins, Def6 and SWAP-70 (Nat Imm 2018). Def6 and SWAP-70 are Rho GEFs that also localize with TFs in the nucleus and have a previously defined role in regulating IRF4 function. The 2018 report defines the chromatin landscape and transcriptome of WT and DKO ABCs and show (via genetic epistasis, RNAseq, and ATACseq) that the SWEF proteins normally repress expansion of ABCs by negatively regulating IRF5. Challenges included isolating effects of germline and conditional B cell deletion strategies to cell-intrinsic roles within ABCs.

In the current manuscript, the authors take a similar set of approaches (genetic epistasis and extensive genome-wide profiling of ABCs and other B cell populations). They begin by observing that autoimmune disease and ABC expansion in this DKO model has a female sex-bias reminiscent of human SLE. They show that TLR7 is required for both ABC expansion and disease in DKO mice, and that adding an extra copy of TLR7 via Yaa overcomes the sex bias in DKO males. They show that disease is dependent on both IRF5 and IRF8 (but only in part on Tbet). Finally, they profile BCRs, transcriptome and epigenome of ABCs and other effector B cell subsets from these animal models. They show that sex bias and TLR7-dose correlates with

an ISG signature in ABCs, and they also identify clonal relatedness between ABCs and effector compartments like GC and PC.

Overall - the role of TLR7 dose in mouse models of SLE has been well-established - dating back to study of Yaa-FcgRIIb disease by the Bolland lab (Pisitkun et al. Science 2007). The requirement of TLR7 for generation of ABCs is also previously shown, and as noted above, contribution of IRF5 to DKO ABC expansion and disease was already established in prior Nat Imm 2018 paper (albeit with CD21cre rather than CD23cre). Finally, the contribution of TLR7 signaling via IRFs to IFN production in context of viral infection and SLE models is also described in the literature. However, the strength of the current report is the mechanistic connection across all of these pathways in the context of an extremely well-characterized and profoundly sex-biased SLE model. The report moreover identifies heterogeneity within effector B cell populations at the level of CD11c expression and associated transcriptomes, and identifies extensive clonal relatedness between ABCs and effector compartments like GC and PC. Importance of dissecting these pathways in the DKO model rests on identification of ABCs as a relevant cell population with pathogenic potential across models and in human disease, as well as genetic associations between IRF5, Def6, ifn pathway, and SLE. For all of these reasons, the current report is of broad interest to researchers in the field of SLE.

The manuscript frames a very extensive and impressive body of work that leans equally on genetic epistasis and transcriptome analyses to develop a mechanistic model for sex-biased disease pathogenesis in DKO animals. The conclusions are supported by the data overall. Queries and comments below are enumerated in detail, but a key question to highlight is the extent to which TLR7 dose is acting in a B cell-intrinsic or ABC-intrinsic manner in this model. Presumably it is to some extent, but there may be a contribution by other type 1 ifn-producing cell populations like pDC that is not addressed. Another question that is hard to address with existing tools is extent to which all machinery studied genetically is operating specifically within ABCs, and extent to which ABCs are indeed the primary pathogenic cell population in this model. While circumstantial evidence supporting role of ABCs is substantial (and increased by the work presented here), these caveats ought to be discussed.

We also thank this Reviewer for the positive comments on our manuscript and truly appreciate all the input and suggestions that the Reviewer offered. We agree with the Reviewer that we cannot definitely rule out the contribution of other cell types, such as pDCs, and that, given the limitations of the systems used, we cannot categorically establish that ABCs are indeed the primary pathogenic population and have now included these caveats in the discussion. We believe, however, that the new epigenetic data that we are providing in Fig.4G-I that show profound differences in the H3K27me3 profiles of ABCs versus FoBs at ISGs further strengthen the idea that the TLR7-mediated effects occur in an ABC-intrinsic manner rather than as a result of an IFN-rich environment (which would be expected to affect both cell compartments) and have included this point in the discussion (pg19).

Below please find a detailed response to the Reviewer's additional comments.

Specific critiques and/or queries:

-Introduction notes "...has recently been shown to escape X-chromosome inactivation" about TLR7. Ref (4) Sci Immunology report from 2018 describes biallelic expression of TLR7 in about 20-30% of several immune cell populations which is striking, but this doesn't imply 2-fold expression of TLR7 broadly. Therefore, I would modify phrase to "...escape complete X-chromosome inactivation" for those readers that aren't familiar with the primary report. Related to this, it is worth highlighting that presumably Yaa male DKO mice have most severe disease

than female DKO mice because x-inactivation is altogether absent with X-segment transposed to Y.

We fully agree with the Reviewer about the findings described in the original Science Immunology report and have now clarified in the Introduction that B cells (and other cells) from females can express higher levels of TLR7 due to partial XCI escape in a proportion of cells. We have also explicitly mentioned in the Discussion (pg.22) that the differences in disease severity between DKO females and Yaa-DKO males could be due to the higher levels of TLR7 expression in Yaa-DKO males resulting from complete rather than partial XCI escape.

-S1a assesses TLR7 transcript in male and female Fo B cells and ABCs. If incomplete x-inactivation is posited to account for higher expression in ABCs, why is this not the case in Fo B cells?

Like the Reviewer, we have also been wondering about the fact that differences in TLR7 expression can be observed in male and female ABCs but not FoBs. We speculate that this may result from the unique and dynamic regulation of XCI that has been described to occur in lymphocytes (Yu et al, 2021) and that the mechanisms underlying this dynamic control may be regulated differently in ABCs versus FoBs. We have now added these comments to the Discussion (pg.22).

Figures 1,2. Evidence for TLR7-dose accounting for sex-specific differences in DKO model is compelling and use of TLR7^{-/-} v. Yaa was an elegant choice. While effects of TLR7 are likely ABC-intrinsic at least in part, this isn't formally demonstrated and since impaired x-inactivation of TLR7 is also reported (cited ref 4) in pDCs, excessive Ifn production by that cell type represents another possible contributing mechanism. Related to this: In Figure 4, the enrichment for ISG signature among female and Yaa ABCs from DKO mice makes sense and could represent TLR7 dose affecting type 1 ifn production by pDCs. It should be noted that impact of TLR7/sex may therefore not be entirely cell-intrinsic per se. Indeed, ISG signature might be enhanced among multiple cell types within these animals.

As mentioned above, we agree with the Reviewer that we cannot definitely rule out the contribution of other cell types, such as pDCs, and have now included this caveat in the discussion (pg.19). We believe, however, that the new epigenetic data that we are providing in Fig.4G-I that show profound differences in the H3K27me3 profiles of ABCs versus FoBs at ISGs further strengthen the idea that the TLR7-mediated effects occur in an ABC-intrinsic manner rather than as a result of an IFN-rich environment (which would be expected to affect both cell compartments) and have included this point in the discussion (pg.19).

Figure 2. Massive spontaneous PC expansion in Yaa DKO males is striking. What isotype does this represent; what do total Ig levels look like? Is this PB/PC expansion exclusively in spl or also in BM (since latter harbors LLPCs, would have different implications about origin of the PCs)?

To address the Reviewer's comments, we are now providing total Ig levels (Fig.S1G) and an evaluation of PB/PC in the BM (Fig. S2B). As can be seen from those data, accumulation of PB/PCs was not observed in BM, a finding similar to what has been reported in other murine lupus models and which could be due both to the preferential expansion of SLPCs (as the Reviewer suggests) as well as to alterations in their ability to migrate to the BM.

Figure 3. Attempt to lineage track formerly Tbet⁺ ABCs with “ZTCE” construct is a very nice complement to clonal relatedness studies. However, looks like Fo B cells can express Tbet too so caveat here is that the origin among ABCs is not definitive. Compelling experiments, but would include this caveat or address it.

We agree with the Reviewer that given that a small fraction of FoBs expresses T-bet in the fate mapping studies, we cannot conclusively establish whether these T-bet⁺ cells are destined to upregulate CD11c and acquire a full-fledged ABC phenotype or whether they represent a separate pool of T-bet⁺ B cells that can also differentiate into such heterogenous progeny and are critical for autoantibody production. We have now added this caveat to the Discussion (pg.20).

Figure 7. Uncoupling ABC accumulation and autoantibody production by B cell-specific Tbet deletion would either argue that ABCs depend on other transcriptional regulators in DKO mice OR it could suggest that ABCs are not the origin and do not drive autoantibody production. Both possibilities should be discussed. For this reason, the Tbet result seems important to present in main figure rather than only in the supplement. This also raises the interesting point of what Tbet expression contributes to the ABC transcriptome. This seems a more significant question than the DEG between CD11c⁺ and CD11c⁻ PB/PCs from Fig 6.

We agree with the Reviewer that this is important information and have now moved these data to the main figure as a new Figure 6 as well as highlighted these caveats in the Discussion (pg.20).

Reviewer #3 (Remarks to the Author):

This is an extensive body of work revealing and then mechanistically understanding the role of TLR7 in sex-specific bias observed in ABCs derived from DEF6 and SWAP70 double knockout mice. The immunologic aspects of this work are solid in my opinion, but I have the following suggestions pertaining the epigenetic work performed.

1) It is understood that Tlr7 can escape X-chromosome inactivation in B cells, but it is not clear if and why this would be augmented in DEF6/SWAP70 DKO mice. It will be of interest to show if there is a difference in X-inactivation in the Tlr7 locus between DKO and wild type mice. This can be done using bisulfite sequencing in the Tlr7 locus where regulatory regions in mice have already been established. In addition, average DNA methylation levels in this locus comparing male and female DKO mice will indirectly reveal if there is evidence of incomplete X-inactivation in female mice specifically in ABCs and not FoBs in keeping with the Tlr7 mRNA expression results provided.

We agree with the Reviewer that gaining a greater understanding of the distinctive features of the epigenetic landscape of ABCs, especially as it pertains to XCI, is important. To address this question as well as gain insights into the potential cell-intrinsic features that might control the acquisition of the ISG signature by ABCs we have opted to investigate the global H3K27me3 profile of these cells by CUT&RUN, which shows selective deactivation of repressive H3K27me3 marks within the intra-TAD boundaries of the Tlr7 gene locus in F-ABCs. These data are now presented in Fig.4I.

2) Additional details should be provided regarding the ATAC-seq experiments. For example, how many million reads were generated per sample? How was tagmentation quality assessed?

We would like to thank the Reviewer for pointing out this oversight and are now providing all these details in Table S1.

3) Using ATAC-seq data, the authors can directly assess transcription factor binding differences, using an analysis such as HINT-ATAC to perform TF footprint analysis if enough reads were generated per sample. This will provide a more direct evidence for the TF arguments made using the enrichment analysis in accessibility peaks.

We greatly appreciate the Reviewer's suggestion. Unfortunately, we have been unable to perform HINT-ATAC since, as the Reviewer mentions, HINT-ATAC requires a greater sequencing depth (~100-200 mln) than what was obtained in our experiments as well as a CHIPseq training set for relevant transcription factors, which should be performed in the relevant cell type (and no such CHIPseq experiments have been performed in ABCs).

REVIEWERS' COMMENTS

Reviewer #1 (Remarks to the Author):

The authors have done a great job in addressing the raised comments.

Reviewer #2 (Remarks to the Author):

The authors have satisfactorily addressed all reviewer queries.

Reviewer #3 (Remarks to the Author):

The authors addressed my comments satisfactorily.

Amr Sawalha